# DEEP LEARNING VIA MESSAGE PASSING ALGORITHMS BASED ON BELIEF PROPAGATION

## ABSTRACT

Message-passing algorithms based on the Belief Propagation (BP) equations constitute a well-known distributed computational scheme. They yield exact marginals on tree-like graphical models and have also proven to be effective in many problems defined on loopy graphs, from inference to optimization, from signal processing to clustering. The BP-based schemes are fundamentally different from stochastic gradient descent (SGD), on which the current success of deep networks is based. In this paper, we present and adapt to mini-batch training on GPUs a family of BP-based message-passing algorithms with a reinforcement term that biases distributions towards locally entropic solutions. These algorithms are capable of training multi-layer neural networks with performance comparable to SGD heuristics in a diverse set of experiments on natural datasets including multi-class image classification and continual learning, while being capable of yielding improved performances on sparse networks. Furthermore, they allow to make approximate Bayesian predictions that have higher accuracy than point-wise ones.

## 1 INTRODUCTION

Belief Propagation is a method for computing marginals and entropies in probabilistic inference problems (Bethe, 1935; Peierls, 1936; Gallager, 1962; Pearl, 1982). These include optimization problems as well once they are written as zero temperature limit of a Gibbs distribution that uses the cost function as energy. Learning is one particular case, in which one wants to minimize a cost which is a data dependent loss function. These problems are generally intractable and message-passing techniques have been particularly successful at providing principled approximations through efficient distributed computations.

A particularly compact representation of inference/optimization problems that is used to build massage-passing algorithms is provided by factor graphs. A factor graph is a bipartite graph composed of variables nodes and factor nodes expressing the interactions among variables. Belief Propagation is exact for tree-like factor graphs (Yedidia et al., 2003)), where the Gibbs distribution is naturally factorized, whereas it is approximate for graphs with loops. Still, loopy BP is routinely used with success in many real world applications ranging from error correcting codes, vision, clustering, just to mention a few. In all these problems, loops are indeed present in the factor graph and yet the variables are weakly correlated at long range and BP gives good results. A field in which BP has a long history is the statistical physics of disordered systems where it is known as Cavity Method (Mézard et al., 1987). It has been used to study the typical properties of spin glass models which represent binary variables interacting through random interactions over a given graph. It is very well known that in spin glass models defined on complete graphs and in locally tree-like random graphs, which are both loopy, the weak correlation conditions between variables may hold and BP give asymptotic exact results (Mézard & Montanari, 2009). Here we will mostly focus on neural networks $\pm 1$ binary weights and sign activation functions, for which the messages and the marginals can be described simply by the difference between the probabilities associated with the +1 and -1 states, the so called *magnetizations*. The effectiveness of BP for deep learning has never been numerically tested in a systematic way, however there is clear evidence that the weak correlation decay condition does not hold and thus BP convergence and approximation quality is unpredictable.

In this paper we explore the effectiveness of a variant of BP that has shown excellent convergence properties in hard optimization problems and in non-convex shallow networks. It goes under the

name of focusing BP (fBP) and is based on a probability distribution, a likelihood, that focuses on highly entropic wide minima, neglecting the contribution to marginals from narrow minima even when they are the majority (and hence dominate the Gibbs distribution). This version of BP is thus expected to give good results only in models that have such wide entropic minima as part of their energy landscape. As discussed in (Baldassi et al., 2016a), a simple way to define fBP is to add a "reinforcement" term to the BP equations: an iteration-dependent local field is introduced for each variable, with an intensity proportional to its marginal probability computed in the previous iteration step. This field is gradually increased until the entire system becomes fully biased on a configuration. The first version of reinforced BP was introduced in (Braunstein & Zecchina, 2006) as a heuristic algorithm to solve the learning problem in shallow binary networks. Baldassi et al. (2016a) showed that this version of BP is a limiting case of fBP, i.e., BP equations written for a likelihood that uses the local entropy function instead of the error (energy) loss function. As discussed in depth in that study, one way to introduce a likelihood that focuses on highly entropic regions is to create $y$ coupled replicas of the original system. fBP equations are obtained as BP equations for the replicated system. It turns out that the fBP equations are identical to the BP equations for the original system with the only addition of a self-reinforcing term in the message passing scheme. The fBP algorithm can be used as a solver by gradually increasing the effect of the reinforcement: one can control the size of the regions over which the fBP equations estimate the marginals by tuning the parameters that appear in the expression of the reinforcement, until the high entropy regions reduce to a single configuration. Interestingly, by keeping the size of the high entropy region fixed, the fBP fixed point allows one to estimate the marginals and entropy relative to the region.

In this work, we present and adapt to GPU computation a family of fBP inspired message passing algorithms that are capable of training multi-layer neural networks with generalization performance and computational speed comparable to SGD. This is the first work that shows that learning by message passing in deep neural networks 1) is possible and 2) is a viable alternative to SGD. Our version of fBP adds the reinforcement term at each mini-batch step in what we call the Posterior-as-Prior (PasP) rule. Furthermore, using the message-passing algorithm not as a solver but as an estimator of marginals allows us to make locally Bayesian predictions, averaging the predictions over the approximate posterior. The resulting generalization error is significantly better than those of the solver, showing that, although approximate, the marginals of the weights estimated by message-passing retain useful information. Consistently with the assumptions underlying fBP, we find that the solutions provided by the message passing algorithms belong to flat entropic regions of the loss landscape and have good performance in continual learning tasks and on sparse networks as well.

We also remark that our PasP update scheme is of independent interest and can be combined with different posterior approximation techniques.

The paper is structured as follows: in Sec. 2 we give a brief review of some related works. In Sec. 3 we provide a detailed description of the message-passing equations and of the high level structure of the algorithms. In Sec. 4 we compare the performance of the message passing algorithms versus SGD based approaches in different learning settings.

## 2 RELATED WORKS

The literature on message passing algorithms is extensive, we refer to Mézard & Montanari (2009) and Zdeborová & Krzakala (2016) for a general overview. More related to our work, multilayer message-passing algorithms have been developed in inference contexts (Manoel et al., 2017; Fletcher et al., 2018), where they have been shown to produce exact marginals under certain statistical assumptions on (unlearned) weight matrices.

The properties of message-passing for learning shallow neural networks have been extensively studied (see Baldassi et al. (2020) and reference therein). Barbier et al. (2019) rigorously show that message passing algorithms in generalized linear models perform asymptotically exact inference under some statistical assumptions. Dictionary learning and matrix factorization are harder problems closely related to deep network learning problems, in particular to the modelling of a single intermediate layer. They have been approached using message passing in Kabashima et al. (2016) and Parker et al. (2014), although the resulting predictions are found to be asymptotically inexact (Maillard et al., 2021). The same problem is faced by the message passing algorithm recently proposed for a multi-layer matrix factorization scenario (Zou et al., 2021). Unfortunately, our framework as well

doesn't yield asymptotic exact predictions. Nonetheless, it gives a message passing heuristic that for the first time is able to train deep neural networks on natural datasets, therefore sets a reference for the algorithmic applications of this research line.

A few papers advocate the success of SGD to the geometrical structure (smoothness and flatness) of the loss landscape in neural networks (Baldassi et al., 2015; Chaudhari et al., 2017; Garipov et al., 2018; Li et al., 2018; Pittorino et al., 2021; Feng & Tu, 2021). These considerations do not depend on the particular form of the SGD dynamics and should extend also to other types of algorithms, although SGD is by far the most popular choice among NNs practitioners due to its simplicity, flexibility, speed, and generalization performance.

While our work focuses on message passing schemes, some of the ideas presented here, such as the PasP rule, can be combined with algorithms for Bayesian neural networks' training (Hernández-Lobato & Adams, 2015; Wu et al., 2018). Recent work extends BP by combining it with graph neural networks (Kuck et al., 2020; Satorras & Welling, 2021). Finally, some work in computational neuroscience shows similarities to our approach (Rao, 2007).

## 3 LEARNING BY MESSAGE PASSING

### 3.1 POSTERIOR-AS-PRIOR UPDATES

We consider a multi-layer perceptron with $L$ hidden neuron layers, having weight and bias parameters $\mathcal{W} = \{\boldsymbol{W}^\ell, \boldsymbol{b}^\ell\}_{\ell=0}^L$. We allow for stochastic activations $P^\ell(\boldsymbol{x}^{\ell+1}|\boldsymbol{z}^\ell)$, where $\boldsymbol{z}^\ell$ is the neuron's pre-activation vector for layer $\ell$, and $P^\ell$ is assumed to be factorized over the neurons. If no stochasticity is present, $P^\ell$ just encodes an element-wise activation function. The probability of output $y$ given an input $\boldsymbol{x}$ is then given by

$$P(y \,|\, \boldsymbol{x}, \mathcal{W}) = \int d\boldsymbol{x}^{1:L} \prod_{\ell=0}^L P^{\ell+1}(\boldsymbol{x}^{\ell+1} \,|\, \boldsymbol{W}^\ell \boldsymbol{x}^\ell + \boldsymbol{b}^\ell), \tag{1}$$

where for convenience we defined $\boldsymbol{x}^0 = \boldsymbol{x}$ and $\boldsymbol{x}^{L+1} = y$. In a Bayesian framework, given a training set $D = \{(\boldsymbol{x}_n, y_n)\}_n$ and a prior distribution over the weights $q_\theta(\mathcal{W})$ in some parametric family, the posterior distribution is given by

$$P(\mathcal{W} \,|\, D, \theta) \propto \prod_n P(y_n \,|\, \boldsymbol{x}_n, \mathcal{W}) \, q_\theta(\mathcal{W}). \tag{2}$$

Here $\propto$ denotes equality up to a normalization factor. Using the posterior one can compute the Bayesian prediction for a new data-point $\boldsymbol{x}$ through the average $P(y \,|\, \boldsymbol{x}, D, \theta) = \int d\mathcal{W} \, P(y \,|\, \boldsymbol{x}, \mathcal{W}) \, P(\mathcal{W} \,|\, D, \theta)$. Unfortunately, the posterior is generically intractable due to the hard-to-compute normalization factor. On the other hand, we are mainly interested in training a distribution that covers wide minima of the loss landscape that generalize well (Baldassi et al., 2016a) and in recovering pointwise estimators within these regions. The Bayesian modeling becomes an auxiliary tool to set the stage for the message passing algorithms seeking flat minima. We also need a formalism that allows for mini-batch training to speed-up the computation and deal with large datasets. Therefore, we devise an update scheme that we call Posterior-as-Prior (PasP), where we evolve the parameters $\theta^t$ of a distribution $q_{\theta^t}(\mathcal{W})$ computed as an approximate mini-batch posterior, in such a way that the outcome of the previous iteration becomes the prior in the following step. In the PasP scheme, $\theta^t$ retains the memory of past observations. We also add an exponential factor $\rho$, that we typically set close to 1, tuning the forgetting rate and playing a role similar to the learning rate in SGD. Given a mini-batch $(\boldsymbol{X}^t, \boldsymbol{y}^t)$ sampled from the training set at time $t$ and a scalar $\rho > 0$, the PasP update reads

$$q_{\theta^{t+1}}(\mathcal{W}) \approx \left[ P(\mathcal{W} \,|\, \boldsymbol{y}^t, \boldsymbol{X}^t, \theta^t) \right]^\rho, \tag{3}$$

where $\approx$ denotes approximate equality and we do not account for the normalization factor. A first approximation may be needed in the computation of the posterior, a second to project the approximate posterior onto the distribution manifold spanned by $\theta$ (Minka, 2001). In practice, we will consider factorized approximate posterior in an exponential family and priors $q_\theta$ in the same family, although Eq. 3 generically allow for more refined approximations.

Notice that setting $\rho = 1$, the batch-size to 1, and taking a single pass over the dataset, we recover the Assumed Density Filtering algorithm (Minka, 2001). For large enough $\rho$ (including $\rho = 1$), the iterations of $q_{\theta^t}$ will concentrate on a pointwise estimator. This mechanism mimics the reinforcement heuristic commonly used to turn Belief Propagation into a solver for constrained satisfaction problems (Braunstein & Zecchina, 2006) and related to flat-minima discovery (see focusing-BP in Baldassi et al. (2016a)). A different prior updating mechanism which can be understood as empirical Bayes has been used in Baldassi et al. (2016b).

## 3.2 INNER MESSAGE PASSING LOOP

While the PasP rule takes care of the reinforcement heuristic across mini-batches, we compute the mini-batch posterior in Eq. 3 using message passing approaches derived from Belief Propagation. BP is an iterative scheme for computing marginals and entropies of statistical models Mézard & Montanari (2009). It is most conveniently expressed on factor graphs, that is bipartite graphs where the two sets of nodes are called variable nodes and factor nodes. They respectively represent the variables involved in the statistical model and their interactions. Message from factor nodes to variable nodes and viceversa are exchanged along the edges of the factor graph for a certain number of BP iterations or until a fixed point is reached.

The factor graph for $P(\mathcal{W} \mid \boldsymbol{X}^t, \boldsymbol{y}^t, \theta^t)$ can be derived from Eq. 2, with the following additional specifications. For simplicity, we will ignore the bias term in each layer. We assume factorized $q_{\theta^t}(\mathcal{W})$, each factor parameterized by its first two moments. In what follows, we drop the PasP iteration index $t$. For each example $(\boldsymbol{x}_n, y_n)$ in the mini-batch, we introduce the auxiliary variables $\boldsymbol{x}_n^\ell, \ell = 1, \ldots, L$, representing the layers' activations. For each example, each neuron in the network contributes a factor node to the factor graph. The scalar components of the weight matrices and the activation vectors become variable nodes. This construction is presented in Appendix A, where we also derive the message update rules on the factor graph. The factor graph thus defined is extremely loopy and straightforward iteration of BP has convergence issues. Moreover, in presence of a homogeneous prior over the weights, the neuron permutation symmetry in each hidden layer induces a strongly attractive symmetric fixed point that hinders learning. We work around these issues by breaking the symmetry at time $t = 0$ with an inhomogeneous prior. In our experiments a little initial heterogeneity is sufficient to obtain specialized neurons at each following time step. Additionally, we do not require message passing convergence in the inner loop (see Algorithm 1) but perform one or a few iterations for each $\theta$ update. We also include an inertia term commonly called damping factor in the message updates (see B.2). As we shall discuss, these simple rules suffice to train deep networks by message passing.

For the inner loop we adapt to deep neural networks four different message passing algorithms, all of which are well known to the literature although derived in simpler settings: Belief Propagation (BP), BP-Inspired (BPI) message passing, mean-field (MF), and approximate message passing (AMP). The last three algorithms can be considered approximations of the first one. In the following paragraphs we will discuss their common traits, present the BP updates as an example, and refer to Appendix A for an in-depth exposition. For all algorithms, message updates can be divided in a forward pass and backward pass, as also done in (Fletcher et al., 2018) in a multi-layer inference setting. The BP algorithm is compactly reported in Algorithm 1.

**Meaning of messages.** All the messages involved in the message passing can be understood in terms of cavity marginals or full marginals (as mentioned in the introduction BP is also known as Cavity Method, see (Mézard & Montanari, 2009)). Of particular relevance are $m_{ki}^\ell$ and $\sigma_{ki}^\ell$, denoting the mean and variance of the weights $W_{ki}^\ell$. The quantities $\hat{x}_{in}^\ell$ and $\Delta_{in}^\ell$ instead denote the mean and variance of the $i$-th neuron's activation in layer $\ell$ for a given input $\boldsymbol{x}_n$.

**Scalar free energies.** All message passing schemes are conveniently expressed in terms of two functions that correspond to the effective free energy (Zdeborová & Krzakala, 2016) of a single

neuron and of a single weight respectively :

$$\varphi^\ell(B, A, \omega, V) = \log \int dx\, dz\, e^{-\frac{1}{2}Ax^2 + Bx} P^\ell(x|z)\, e^{-\frac{(\omega - z)^2}{2V}} \qquad \ell = 1, \dots, L \qquad (4)$$

$$\psi(H, G, \theta) = \log \int dw\, e^{-\frac{1}{2}G^2 w^2 + Hw} q_\theta(w) \qquad (5)$$

Notice that for common deterministic activations such as ReLU and sign, the function $\varphi$ has analytic and smooth expressions (see Appendix A.8). The same holds for the function $\psi$ when $q_\theta(w)$ is Gaussian (continuous weights) or a mixture of atoms (discrete weights). At the last layer we impose $P^{L+1}(y|z) = \mathbb{I}(y = \operatorname{sign}(z))$ in binary classification tasks and $P^{L+1}(y|z) = \mathbb{I}(y = \arg\max(z))$ in multi-class classification (see Appendix A.9). While in our experiments we use hard constraints for the final output, therefore solving a constraint satisfaction problem, it would be interesting to also consider soft constraints and introduce a temperature, but this is beyond the scope of our work.

**Start and end of message passing.** At the beginning of a new PasP iteration $t$, we reset the messages (see Appendix A) and run message passing for $\tau_{\max}$ iterations. We then compute the new prior's parameters $\theta^{t+1}$ from the posterior given by the message passing.

**BP Forward pass.** After initialization of the messages at time $\tau = 0$, for each following time we propagate a set of message from the first to the last layer and then another set from the last to the first. For an intermediate layer $\ell$ the forward pass reads

$$\hat{x}_{in \to k}^{\ell, \tau} = \partial_B \varphi^\ell \left( B_{in \to k}^{\ell, \tau-1}, A_{in}^{\ell, \tau-1}, \omega_{in}^{\ell-1, \tau}, V_{in}^{\ell-1, \tau} \right) \qquad (6)$$

$$\Delta_{in}^{\ell, \tau} = \partial_B^2 \varphi^\ell \left( B_{in}^{\ell, \tau-1}, A_{in}^{\ell, \tau-1}, \omega_{in}^{\ell-1, \tau}, V_{in}^{\ell-1, \tau} \right) \qquad (7)$$

$$m_{ki \to n}^{\ell, \tau} = \partial_H \psi(H_{ki \to n}^{\ell, \tau-1}, G_{ki}^{\ell, \tau-1}, \theta_{ki}^\ell) \qquad (8)$$

$$\sigma_{ki}^{\ell, \tau} = \partial_H^2 \psi(H_{ki}^{\ell, \tau-1}, G_{ki}^{\ell, \tau-1}, \theta_{ki}^\ell) \qquad (9)$$

$$V_{kn}^{\ell, \tau} = \sum_i \left( \left( m_{ki \to n}^{\ell, \tau} \right)^2 \Delta_{in}^{\ell, \tau} + \sigma_{ki}^{\ell, \tau} (\hat{x}_{in \to k}^{\ell, \tau})^2 + \sigma_{ki}^{\ell, \tau} \Delta_{in}^{\ell, \tau} \right) \qquad (10)$$

$$\omega_{kn \to i}^{\ell, \tau} = \sum_{i' \neq i} m_{ki' \to n}^{\ell, \tau} \hat{x}_{i'n \to k}^{\ell, \tau} \qquad (11)$$

The equations for the first layer differ slightly and in an intuitive way from the ones above (see Appendix A.3).

**BP Backward pass.** The backward pass updates a set of messages from the last to the first layer:

$$g_{kn \to i}^{\ell, \tau} = \partial_\omega \varphi^{\ell+1} \left( B_{kn}^{\ell+1, \tau}, A_{kn}^{\ell+1, \tau}, \omega_{kn \to i}^{\ell, \tau}, V_{kn}^{\ell, \tau} \right) \qquad (12)$$

$$\Gamma_{kn}^{\ell, \tau} = -\partial_\omega^2 \varphi^{\ell+1} \left( B_{kn}^{\ell+1, \tau}, A_{kn}^{\ell+1, \tau}, \omega_{kn}^{\ell, \tau}, V_{kn}^{\ell, \tau} \right) \qquad (13)$$

$$A_{in}^{\ell, \tau} = \sum_k \left( (m_{ki \to n}^{\ell, \tau})^2 + \sigma_{ki}^{\ell, \tau} \right) \Gamma_{kn}^{\ell, \tau} - \sigma_{ki}^{\ell, \tau} \left( g_{kn \to i}^{\ell, \tau} \right)^2 \qquad (14)$$

$$B_{in \to k}^{\ell, \tau} = \sum_{k' \neq k} m_{k'i \to n}^{\ell, \tau} g_{k'n \to i}^{\ell, \tau} \qquad (15)$$

$$G_{ki}^{\ell, \tau} = \sum_n \left( (\hat{x}_{in \to k}^{\ell, \tau})^2 + \Delta_{in}^{\ell, \tau} \right) \Gamma_{kn}^{\ell, \tau} - \Delta_{in}^{\ell, \tau} \left( g_{kn \to i}^{\ell, \tau} \right)^2 \qquad (16)$$

$$H_{ki \to n}^{\ell, \tau} = \sum_{n' \neq n} \hat{x}_{in' \to k}^{\ell, \tau} g_{kn' \to i}^{\ell, \tau} \qquad (17)$$

As with the forward pass, we add the caveat that for the last layer the equations are slightly different from the ones above.

**Computational complexity** The message passing equations boil down to element-wise operations and tensor contractions that we easily implement using the GPU friendly julia library Tullio.jl (Abbott et al., 2021). For a layer of input and output size $N$ and considering a batch-size of $B$, the time complexity of a forth-and-back iteration is $O(N^2 B)$ for all message passing algorithms (BP, BPI, MF, and AMP), the same as SGD. The prefactor varies and it is generally larger than SGD (see Appendix B.9). Also, time complexity for message passing is proportional to $\tau_{\max}$ (which we typically set to 1). We provide our implementation in the GitHub repo **anonymous**.

---

**Algorithm 1:** BP for deep neural networks

---

```
// Message passing used in the PasP Eq.  3 to approximate.
// the mini-batch posterior.
// Here we specifically refer to BP updates.
// BPI, MF, and AMP updates take the same form but using
// the rules in Appendix A.4, A.5, and A.7 respectively
```
1  Initialize messages.
2  **for** $\tau = 1, \ldots \tau_{\max}$ **do**
       `// Forward Pass`
3      **for** $l = 0, \ldots, L$ **do**
4          compute $\hat{\boldsymbol{x}}^\ell, \boldsymbol{\Delta}^\ell$ using (6, 7)
5          compute $\boldsymbol{m}^\ell, \boldsymbol{\sigma}^\ell$ using (8, 9)
6          compute $\mathbf{V}^\ell, \boldsymbol{\omega}^\ell$ using (10, 11)
       `// Backward Pass`
7      **for** $l = L, \ldots, 0$ **do**
8          compute $\boldsymbol{g}^\ell, \boldsymbol{\Gamma}^\ell$ using (12, 13)
9          compute $\boldsymbol{A}^\ell, \boldsymbol{B}^\ell$ using (14, 15)
10         compute $\boldsymbol{G}^\ell, \boldsymbol{H}^\ell$ using (16, 17)

---

## 4 NUMERICAL RESULTS

We implement our message passing algorithms on neural networks with continuous and binary weights and with binary activations. In our experiments we fix $\tau_{\max} = 1$. We typically do not observe an increase in performance taking more steps, except for some specific cases and in particular for MF layers. We remark that for $\tau_{\max} = 1$ the BP and the BPI equations are identical, so in most of the subsequent numerical results we will only investigate BP.

We compare our algorithms with a SGD-based algorithm adapted to binary architectures (Hubara et al., 2016) which we call BinaryNet along the paper (see Appendix B.6 for details). Comparison of Bayesian predictions are with the gradient-based Expectation Backpropagation (EBP) algorithm (Soudry et al., 2014a), also able to deal with discrete weights and activations. In all architectures we avoid the use of bias terms and batch-normalization layers.

We find that message-passing algorithms are able to train generic MLP architectures with varying numbers and sizes of hidden layers. As for the datasets, we are able to perform both binary classification and multi-class classification on standard computer vision datasets such as MNIST, Fashion-MNIST, and CIFAR-10. Since these datasets consist of 10 classes, for the binary classification task we divide each dataset in two classes (even vs odd).

We report that message passing algorithms are able to solve these optimization problems with generalization performance comparable to or better than SGD-based algorithms. Some of the message passing algorithms (BP and AMP in particular) need fewer epochs to achieve low error than the ones required by SGD-based algorithms, even if adaptive methods like Adam are considered. Timings of our GPU implementations of message passing algorithms are competitive with SGD (see Appendix B.9).

### 4.1 EXPERIMENTS ACROSS ARCHITECTURES

We select a specific task, multi-class classification on Fashion-MNIST, and we compare the message passing algorithms with BinaryNet for different choices of the architecture (i.e. we vary the number and the size of the hidden layers). In Fig.1 (Left) we present the learning curves for a MLP with 3 hidden layers with 501 units with binary weights and activations. Similar results hold in our experiments with 2 or 3 hidden layers of 101, 501 or 1001 units and with batch sizes from 1 to from 1024. The parameters used in our simulations are reported in Appendix B.3. Results on networks with continuous weights can be found in Fig.2 (Right).

### 4.2 SPARSE LAYERS

Since the BP algorithm has notoriously been successful on sparse graphs, we perform a straight-forward implementation of pruning at initialization, i.e. we impose a random boolean mask on the weights that we keep fixed along the training. We call *sparsity* the fraction of zeroed weights. This kind of non-adaptive pruning is known to largely hinder learning (Frankle et al., 2021; Sung et al., 2021). In the right panel of Fig. 1, we report results on sparse binary networks in which we train a MLP with 2 hidden layers of 101 units on the MNIST dataset. For reference, results on pruning quantized/binary networks can be found in Refs. (Han et al., 2016; Ardakani et al., 2017; Tung & Mori, 2018; Diffenderfer & Kailkhura, 2021). Experimenting with sparsity up to 90%, we observe that BP and MF perform better than BinaryNet. AMP struggles behind BinaryNet instead.

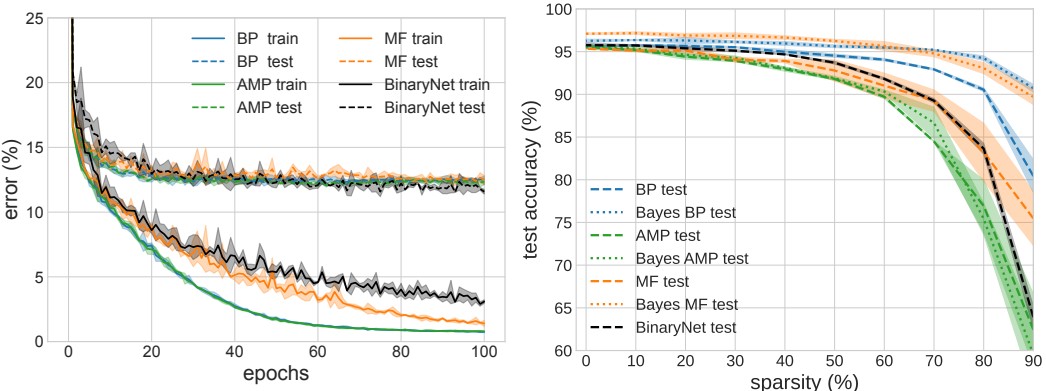

Figure 1: (Left) Training curves of message passing algorithms compared with BinaryNet on the Fashion-MNIST dataset (multi-class classification) with a binary MLP with 3 hidden layers of 501 units. (Right) Final test accuracy when varying the layer's sparsity in a binary MLP with 2 hidden layers of 101 units on the MNIST dataset (multi-class). In both panels the batch-size is 128 and curves are averaged over 5 realizations of the initial conditions (and sparsity pattern in the right panel).

### 4.3 EXPERIMENTS ACROSS DATASETS

We now fix the architecture, a MLP with 2 hidden layers of 501 neurons each with binary weights and activations. We vary the dataset, i.e. we test the BP-based algorithms on standard computer vision benchmark datasets such as MNIST, Fashion-MNIST and CIFAR-10, in both the multi-class and binary classification tasks. In Tab. 1 we report the final test errors obtained by the message passing algorithms compared to the BinaryNet baseline. See Appendix B.4 for the corresponding training errors and the parameters used in the simulations. We mention that while the test performance is mostly comparable, the train error tends to be lower for the message passing algorithms.

| Dataset | BinaryNet | BP | AMP | MF |
|---|---|---|---|---|
| **MNIST (2 classes)** | $1.3 \pm 0.1$ | $1.4 \pm 0.2$ | $1.4 \pm 0.1$ | $1.3 \pm 0.2$ |
| **Fashion-MNIST (2 classes)** | $2.4 \pm 0.1$ | $2.3 \pm 0.1$ | $2.4 \pm 0.1$ | $2.3 \pm 0.1$ |
| **CIFAR-10 (2 classes)** | $30.0 \pm 0.3$ | $31.4 \pm 0.1$ | $31.1 \pm 0.3$ | $31.1 \pm 0.4$ |
| **MNIST** | $2.2 \pm 0.1$ | $2.6 \pm 0.1$ | $2.6 \pm 0.1$ | $2.3 \pm 0.1$ |
| **Fashion-MNIST** | $12.0 \pm 0.6$ | $11.8 \pm 0.3$ | $11.9 \pm 0.2$ | $12.1 \pm 0.2$ |
| **CIFAR-10** | $59.0 \pm 0.7$ | $58.7 \pm 0.3$ | $58.5 \pm 0.2$ | $60.4 \pm 1.1$ |

Table 1: Test error (%) on Fashion-MNIST of various algorithms on a MLP with 2 hidden layers of 501 units, binary weights and activations. All algorithms are trained with batch-size 128 and for 100 epochs. Mean and standard deviations are calculated over 5 random initializations.

### 4.4 LOCALLY BAYESIAN ERROR

The message passing framework used as an estimator of the mini-batch posterior marginals allows us to perform approximate Bayesian prediction, i.e. averaging the pointwise predictions over the approximate posterior. We observe better generalization error from Bayesian predictions compared to point-wise ones, showing that the marginals retain useful information. However, we roughly estimate the marginals with the PasP mini-batch procedure (the exact ones should be computed with a full-batch procedure, but this converges with difficulty in our tests). Since BP-based algorithms tend to focus on dense states (as also confirmed by the local energy measure performed in Appendix B.5), the Bayesian error we compute can be considered as a local approximation of the full one. We report results for binary classification on the MNIST dataset in Fig. 2, and we observe the same performance increase on different datasets and architectures. We obtain the Bayesian prediction from the output marginal given by a single forward pass of the message passing. To obtain good Bayesian estimates it is important that the posterior distribution does not concentrate too much, otherwise the Bayesian prediction will converge to the prediction of a single configuration.

In Fig.2 we also perform a comparison of BP (point-wise and Bayesian) with SGD and another algorithm able to perform Bayesian predictions, Expectation Backpropagation (Soudry et al., 2014a) see Appendix B.7 for implementation details.

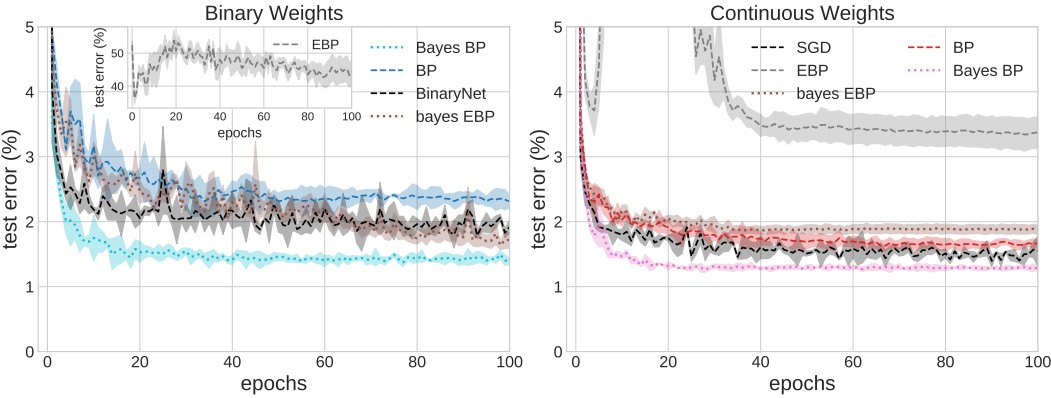

Figure 2: (Left) Test error curves for Bayesian and point-wise predictions for a MLP with 2 hidden layers of 101 units on the 2-classes MNIST dataset. We report the results for (Left) binary and (Right) continuous weights. In both cases, we compare SGD, BP (point-wise and Bayesian) and EBP (point-wise and Bayesian). See Appendix B.3 for details.

### 4.5 CONTINUAL LEARNING

Given the high local entropy (i.e. the flatness) of the solutions found by the BP-based algorithms (see Appendix B.5), we perform additional tests in a classic setting, continual learning, where the

possibility of locally rearranging the solutions while keeping low training error can be an advantage. When a deep network is trained sequentially on different tasks, it tends to forget exponentially fast previously seen tasks while learning new ones (McCloskey & Cohen, 1989; Robins, 1995; Fusi et al., 2005). Recent work (Feng & Tu, 2021) has shown that searching for a flat region in the loss landscape can indeed help to prevent catastrophic forgetting. Several heuristics have been proposed to mitigate the problem (Kirkpatrick et al., 2017; Aljundi et al., 2018; Zenke et al., 2017; Laborieux et al., 2021) but all require specialized adjustments to the loss or the dynamics .

Here we show instead that our message passing schemes are naturally prone to learn multiple tasks sequentially, mitigating the characteristic memory issues of the gradient-based schemes without the need for explicit modifications. As a prototypical experiment, we sequentially trained a multi-layer neural network on 6 different versions of the MNIST dataset, where the pixels of the images have been randomly permuted (Goodfellow et al., 2013), giving a fixed budget of 40 epochs on each task. We present the results for a two hidden layer neural network with 2001 units on each layer (see Appendix B.3 for details). As can be seen in Fig. 3, at the end of the training the BP algorithm is able to reach good generalization performances on all the tasks. We compared the BP performance with BinaryNet, which already performs better than SGD with continuous weights (see the discussion in Laborieux et al. (2021)). While our BP implementation is not competitive with ad-hoc techniques specifically designed for this problem, it beats non-specialized heuristics. Moreover, we believe that specialized approaches like the one of Laborieux et al. (2021) can be adapted to message passing as well.

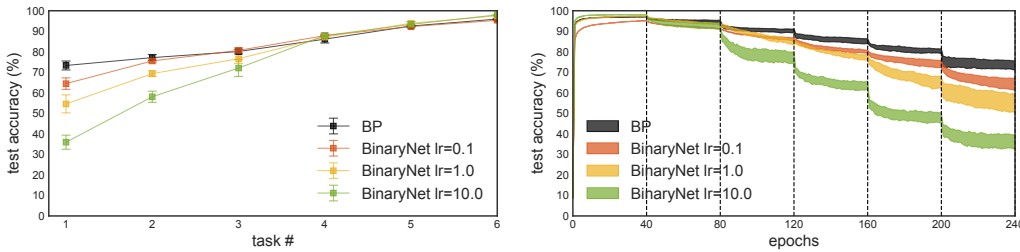

Figure 3: Performance of BP and BinaryNet on the permuted MNIST task (see text) for a two hidden layer network with 2001 units on each layer and binary weights and activations. The model is trained sequentially on 6 different versions of the MNIST dataset (the tasks), where the pixels have been permuted. (Left) Test accuracy on each task after the network has been trained on all the tasks. (Right) Test accuracy on the first task as a function of the number of epochs. Points are averages over 5 independent runs, shaded areas are errors on the mean.

## 5 DISCUSSION AND CONCLUSIONS

While successful in many fields, message passing algorithms, have notoriously struggled to scale to deep neural networks training problems. Here we have developed a class of fBP-based message passing algorithms and used them within an update scheme, Posterior-as-Prior (PasP), that makes it possible to train deep and wide multilayer perceptrons by message passing.

We performed experiments binary activations and either binary or continuous weights. Future work should try to include different activations, biases, batch-normalization, and convolutional layers as well. Another interesting direction is the algorithmic computation of the (local) entropy of the model from the messages.

Further theoretical work is needed for a more complete understanding of the robustness of our methods. Recent developments in message passing algorithms (Rangan et al., 2019) and related theoretical analysis (Goldt et al., 2020) could provide fruitful inspirations. While our algorithms can be used for approximate Bayesian inference, exact posterior calculation is still out of reach for message passing approaches and much technical work is needed in that direction.

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

# Appendices

## Contents

## A   BP-BASED MESSAGE PASSING ALGORITHMS

### A.1   PRELIMINARY CONSIDERATIONS

Given a mini-batch $\mathcal{B} = \{(\boldsymbol{x}_n, y_n)\}_n$, the factor graph defined by Eqs. (1, 2, 18) is explicitly written as:

$$P(\mathcal{W}, \boldsymbol{x}^{1:L} \,|\, \mathcal{B}, \theta) \propto \prod_{\ell=0}^{L} \prod_{k,n} P^{\ell+1}\left(x_{kn}^{\ell+1} \,\bigg|\, \sum_i W_{ki}^\ell x_{in}^\ell\right) \prod_{k,i,\ell} q_\theta(W_{ki}^\ell), \qquad (18)$$

where $\boldsymbol{x}_n^0 = \boldsymbol{x}_n$, $\boldsymbol{x}_n^{L+1} = y_n$. The derivation of the BP equations for this model is straightforward albeit lengthy and involved. It is obtained following the steps presented in multiple papers, books, and reviews, see for instance (Mézard & Montanari, 2009; Zdeborová & Krzakala, 2016; Mézard, 2017), although it has not been attempted before in deep neural networks. It should be noted that a (common) approximation that we take here with respect to the standard BP scheme, is that messages are assumed to be Gaussian distributed and therefore parameterized by their mean and variance. This goes by the name of relaxed belied propagation (rBP), just referred to as BP throughout the paper.

We derive the BP equations in A.2 and present them all together in A.3. From BP, we derive other 3 message passing algorithms useful for the deep network training setting, all of which are well known to the literature: BP-Inspired (BPI) message passing A.4, mean-field (MF) A.5, and approximate

message passing (AMP) A.7. The AMP derivation is the more involved and given in A.6. In all these cases, message updates can be divided in a forward pass and a backward pass, as also done in Fletcher et al. (2018) in a multi-layer inference setting. The BP algorithm is compactly reported in Algorithm 1.

In our notation, $\ell$ denotes the layer index, $\tau$ the BP iteration index, $k$ an output neuron index, $i$ an input neuron index, and $n$ a sample index.

We report below, for convenience, some of the considerations also present in the main text.

**Meaning of messages.** All the messages involved in the message passing equations can be understood in terms of cavity marginals or full marginals (as mentioned in the introduction BP is also known as the Cavity Method, see Mézard & Montanari (2009)). Of particular relevance are the quantities $m_{ki}^\ell$ and $\sigma_{ki}^\ell$, denoting the mean and variance of the weights $W_{ki}^\ell$. The quantities $\hat{x}_{in}^\ell$ and $\Delta_{in}^\ell$ instead denote mean and variance of the $i$-th neuron's activation in layer $\ell$ in correspondence of an input $\boldsymbol{x}_n$.

**Scalar free energies.** All message passing schemes can be expressed using the following scalar functions, corresponding to single neuron and single weight effective free-energies respectively:

$$\varphi^\ell(B, A, \omega, V) = \log \int \mathrm{d}x \, \mathrm{d}z \, e^{-\frac{1}{2}Ax^2 + Bx} \, P^\ell(x \,|\, z) \, e^{-\frac{(\omega-z)^2}{2V}}, \tag{19}$$

$$\psi(H, G, \theta) = \log \int \mathrm{d}w \, e^{-\frac{1}{2}G^2 w^2 + Hw} \, q_\theta(w). \tag{20}$$

These free energies will naturally arise in the derivation of the BP equations in Appendix A.2. For the last layer, the neuron function has to be slightly modified:

$$\varphi^{L+1}(y, \omega, V) = \log \int \mathrm{d}z \, P^{L+1}(y \,|\, z) \, e^{-\frac{(\omega-z)^2}{2V}}. \tag{21}$$

Notice that for common deterministic activations such as ReLU and $\mathrm{sign}$, the function $\varphi$ has analytic and smooth expressions that we give in Appendix A.8. Same goes for $\psi$ when $q_\theta(w)$ is Gaussian (continuous weights) or a mixture of atoms (discrete weights). At the last layer we impose $P^{L+1}(y|z) = \mathbb{I}(y = \mathrm{sign}(z))$ in binary classification tasks. For multi-class classification instead, we have to adapt the formalism to vectorial pre-activations $\boldsymbol{z}$ and assume $P^{L+1}(y|\boldsymbol{z}) = \mathbb{I}(y = \arg\max(\boldsymbol{z}))$ (see Appendix A.9). While in our experiments we use hard constraints for the final output, therefore solving a constraint satisfaction problem, it would be interesting to also consider generic loss functions. That would require minimal changes to our formalism, but this is beyond the scope of our work.

**Binary weights.** In our experiments we use $\pm 1$ weights in each layer. Therefore each marginal can be parameterized by a single number and our prior/posterior takes the form

$$q_\theta(W_{ki}^\ell) \propto e^{\theta_{ki}^\ell W_{ki}^\ell} \tag{22}$$

The effective free energy function Eq. 20 becomes

$$\psi(H, G, \theta_{ki}^\ell) = \log 2 \cosh(H + \theta_{ki}^\ell) \tag{23}$$

and the messages $G$ can be dropped from the message passing.

**Start and end of message passing.** At the beginning of a new PasP iteration $t$, we reset the messages to zero and run message passing for $\tau_{\max}$ iterations. We then compute the new prior $q_{\theta^{t+1}}(\mathcal{W})$ from the posterior given by the message passing iterations.

## A.2 DERIVATION OF THE BP EQUATIONS

In order to derive the BP equations, we start with the following portion of the factor graph reported in Eq. 18 in the main text, describing the contribution of a single data example in the inner loop of the PasP updates:

$$\prod_{\ell=0}^{L} \prod_{k} P^{\ell+1} \left( x_{kn}^{\ell+1} \,\middle|\, \sum_{i} W_{ki}^{\ell} x_{in}^{\ell} \right) \quad \text{where } \boldsymbol{x}_n^0 = \boldsymbol{x}_n, \ \boldsymbol{x}_n^{L+1} = y_n. \tag{24}$$

where we recall that the quantity $x_{kn}^{\ell}$ corresponds to the activation of neuron $k$ in layer $\ell$ in correspondence of the input example $n$.

Let us start by analyzing the single factor:

$$P^{\ell+1} \left( x_{kn}^{\ell+1} \,\middle|\, \sum_{i} W_{ki}^{\ell} x_{in}^{\ell} \right) \tag{25}$$

We refer to messages that travel from input to output in the factor graph as *upgoing* or *upwards* messages, while to the ones that travel from output to input as *downgoing* or *backwards* messages.

**Factor-to-variable-W messages**   The factor-to-variable-$W$ messages read:

$$\hat{\nu}_{kn\to ki}^{\ell+1}(W_{ki}^{\ell}) \propto \int \prod_{i'\neq i} d\nu_{ki'\to n}^{\ell}(W_{ki'}^{\ell}) \prod_{i'} d\nu_{i'n\to k}^{\ell}(x_{i'n}^{\ell}) \, d\nu_{\downarrow}(x_{kn}^{\ell+1}) \, P^{\ell+1} \left( x_{kn}^{\ell+1} \,\middle|\, \sum_{i'} W_{ki'}^{\ell} x_{i'n}^{\ell} \right) \tag{26}$$

where $\nu_{\downarrow}$ denotes the messages travelling downwards (from output to input) in the factor graph.

We denote the means and variances of the incoming messages respectively with $m_{ki\to n}^{\ell}$, $\hat{x}_{in\to k}^{\ell}$ and $\sigma_{ki\to n}^{\ell}$, $\Delta_{in\to k}^{\ell}$:

$$m_{ki\to n}^{\ell} = \int d\nu_{ki\to n}^{\ell}(W_{ki}^{\ell}) \, W_{ki}^{\ell} \tag{27}$$

$$\sigma_{ki\to n}^{\ell} = \int d\nu_{ki\to n}^{\ell}(W_{ki}^{\ell}) \, \left( W_{ki}^{\ell} - m_{ki\to n}^{\ell} \right)^2 \tag{28}$$

$$\hat{x}_{in\to k}^{\ell} = \int d\nu_{in\to k}^{\ell}(x_{in}^{\ell}) \, x_{in}^{\ell} \tag{29}$$

$$\Delta_{in\to k}^{\ell} = \int d\nu_{in\to k}^{\ell}(x_{in}^{\ell}) \, \left( x_{in}^{\ell} - \hat{x}_{in\to k}^{\ell} \right)^2 \tag{30}$$

We now use the central limit theorem to observe that with respect to the incoming messages distributions - assuming independence of these messages - in the large input limit the preactivation is a Gaussian random variable:

$$\sum_{i'\neq i} W_{ki'}^{\ell} x_{i'n}^{\ell} \sim \mathcal{N}(\omega_{kn\to i}^{\ell}, V_{kn\to i}^{\ell}) \tag{31}$$

where:

$$\omega_{kn\to i}^{\ell} = \mathbb{E}_{\nu} \left[ \sum_{i'\neq i} W_{ki'}^{\ell} x_{i'n}^{\ell} \right] = \sum_{i'\neq i} m_{ki'\to n}^{\ell} \hat{x}_{i'n\to k}^{\ell} \tag{32}$$

$$
\begin{aligned}
V_{kn\to i}^{\ell} &= Var_{\nu} \left[ \sum_{i'\neq i} W_{ki'}^{\ell} x_{i'n}^{\ell} \right] \\
&= \sum_{i'\neq i} \left( \sigma_{ki'\to n}^{\ell} \Delta_{i'n\to k}^{\ell} + \left( m_{ki'\to n}^{\ell} \right)^2 \Delta_{i'n\to k}^{\ell} + \sigma_{ki'\to n}^{\ell} \left( \hat{x}_{i'n\to k}^{\ell} \right)^2 \right)
\end{aligned} \tag{33}
$$

Therefore we can rewrite the outgoing messages as:

$$\hat{\nu}_{kn\rightarrow i}^{\ell+1}(W_{ki}^{\ell}) \propto \int dz \, d\nu_{in\rightarrow k}^{\ell}(x_{in}^{\ell}) \, d\nu_{\downarrow}(x_{kn}^{\ell+1}) \, e^{-\frac{\left(z-\omega_{kn\rightarrow i}-W_{ki}^{\ell}x_{in}^{\ell}\right)^2}{2V_{kn\rightarrow i}}} P^{\ell+1}\left(x_{kn}^{\ell+1} \,\middle|\, z\right) \qquad (34)$$

We now assume $W_{ki}^{\ell}x_{in}^{\ell}$ to be small compared to the other terms. With a second order Taylor expansion we obtain:

$$\hat{\nu}_{kn\rightarrow i}^{\ell}(W_{ki}^{\ell}) \propto \int dz \, d\nu_{\downarrow}(x_{kn}^{\ell+1}) \, e^{-\frac{(z-\omega_{kn\rightarrow i})^2}{2V_{kn\rightarrow i}}} P^{\ell+1}\left(x_{kn}^{\ell+1} \,\middle|\, z\right)$$
$$\times \left(1 + \frac{z - \omega_{kn\rightarrow i}}{V_{kn\rightarrow i}}\hat{x}_{in\rightarrow k}^{\ell}W_{ki}^{\ell} + \frac{(z-\omega_{kn\rightarrow i})^2 - V_{kn\rightarrow i}}{2V_{kn\rightarrow i}}\left(\Delta + \left(\hat{x}_{in\rightarrow k}^{\ell}\right)^2\right)\left(W_{ki}^{\ell}\right)^2\right)$$
$$(35)$$

Introducing now the function:

$$\varphi^{\ell}(B, A, \omega, V) = \log \int dx \, dz \, e^{-\frac{1}{2}Ax^2 + Bx} \, P^{\ell}\left(x|z\right) e^{-\frac{(\omega-z)^2}{2V}} \qquad (36)$$

and defining:

$$g_{kn\rightarrow i}^{\ell} = \partial_{\omega}\varphi^{\ell+1}(B^{\ell+1}, A^{\ell+1}, \omega_{kn\rightarrow i}^{\ell}, V_{kn\rightarrow i}^{\ell}) \qquad (37)$$
$$\Gamma_{kn\rightarrow i}^{\ell} = -\partial_{\omega}^2\varphi^{\ell+1}(B^{\ell+1}, A^{\ell+1}, \omega_{kn\rightarrow i}^{\ell}, V_{kn\rightarrow i}^{\ell}) \qquad (38)$$

the expansion for the log-message reads:

$$\log \hat{\nu}_{kn\rightarrow i}^{\ell}(W_{ki}^{\ell}) \approx const + \hat{x}_{in\rightarrow k}^{\ell} \, g_{kn\rightarrow i}^{\ell} W_{ki}^{\ell}$$
$$- \frac{1}{2}\left(\left(\Delta_{in\rightarrow k}^{\ell} + \left(\hat{x}_{in\rightarrow k}^{\ell}\right)^2\right)\Gamma_{kn\rightarrow i}^{\ell} - \Delta_{in\rightarrow k}^{\ell}\left(g_{kn\rightarrow i}^{\ell}\right)^2\right)\left(W_{ki}^{\ell}\right)^2 \qquad (39)$$

**Factor-to-variable-x messages** The derivation of these messages is analogous to the factor-to-variable-$W$ ones in Eq. 26 just reported. The final result for the log-message is:

$$\log \hat{\nu}_{kn\rightarrow i}^{\ell}(x_{in}^{\ell}) \approx const + m_{ki\rightarrow n}^{\ell} \, g_{kn\rightarrow i}^{\ell} x_{in}^{\ell}$$
$$- \frac{1}{2}\left(\left(\sigma_{ki\rightarrow n}^{\ell} + \left(m_{ki\rightarrow n}^{\ell}\right)^2\right)\Gamma_{kn\rightarrow i}^{\ell} - \sigma_{ki\rightarrow n}^{\ell}\left(g_{kn\rightarrow i}^{\ell}\right)^2\right)\left(x_{in}^{\ell}\right)^2 \qquad (40)$$

**Variable-W-to-output-factor messages** The message from variable $W_{ki}^{\ell}$ to the output factor $kn$ reads:

$$\nu_{ki\rightarrow n}^{\ell}(W_{ki}^{\ell}) \propto P_{\theta_{ki}}^{\ell}(W_{ki}^{\ell})e^{\sum_{n'\neq n}\log\hat{\nu}_{kn'\rightarrow i}^{\ell}(W_{ki}^{\ell})}$$
$$\approx P_{\theta_{ki}}^{\ell}(W_{ki}^{\ell})e^{H_{ki\rightarrow n}^{\ell}W_{ki}^{\ell} - \frac{1}{2}G_{ki\rightarrow n}^{\ell}\left(W_{ki}^{\ell}\right)^2} \qquad (41)$$

where we have defined:

$$H_{ki\rightarrow n}^{\ell} = \sum_{n'\neq n} \hat{x}_{in'\rightarrow k}^{\ell} \, g_{kn'\rightarrow i}^{\ell} \qquad (42)$$
$$G_{ki\rightarrow n}^{\ell} = \sum_{n'\neq n}\left(\left(\Delta_{in'\rightarrow k}^{\ell} + \left(\hat{x}_{in'\rightarrow k}^{\ell}\right)^2\right)\Gamma_{kn'\rightarrow i}^{\ell} - \Delta_{in'\rightarrow k}^{\ell}\left(g_{kn'\rightarrow i}^{\ell}\right)^2\right) \qquad (43)$$

Introducing now the effective free energy:

$$\psi(H, G, \theta) = \log \int dW \ P_\theta^\ell(W) \, e^{HW - \frac{1}{2}GW^2} \tag{44}$$

we can express the first two cumulants of the message $\nu_{ki \to n}^\ell(W_{ki}^\ell)$ as:

$$m_{ki \to n}^\ell = \partial_H \psi(H_{ki \to n}^\ell, G_{ki \to n}^\ell, \theta_{ki}) \tag{45}$$

$$\sigma_{ki \to n}^\ell = \partial_H^2 \psi(H_{ki \to n}^\ell, G_{ki \to n}^\ell, \theta_{ki}) \tag{46}$$

**Variable-x-to-input-factor messages**   We can write the downgoing message as:

$$\nu_\downarrow(x_{in}^\ell) \propto e^{\sum_k \log \hat{\nu}_{kn \to i}^\ell(x_{in}^\ell)}$$
$$\approx e^{B_{in}^\ell x - \frac{1}{2}A_{in}^\ell x^2} \tag{47}$$

where:

$$B_{in}^\ell = \sum_n m_{ki \to n}^\ell \, g_{kn \to i}^\ell \tag{48}$$

$$A_{in}^\ell = \sum_n \left( \left( \sigma_{ki \to n}^\ell + \left( m_{ki \to n}^\ell \right)^2 \right) \Gamma_{kn \to i}^\ell - \sigma_{ki \to n}^\ell \left( g_{kn \to i}^{\ell+1} \right)^2 \right) \tag{49}$$

**Variable-x-to-output-factor messages**   By defining the following cavity quantities:

$$B_{in \to k}^\ell = B_{in \to k}^\ell - m_{ki \to n}^\ell \, g_{kn \to i}^\ell \tag{50}$$

$$A_{in \to k}^\ell = A_{in \to k}^\ell - \left( \left( \sigma_{ki \to n}^\ell + \left( m_{ki \to n}^\ell \right)^2 \right) \Gamma_{kn \to i}^\ell - \sigma_{ki \to n}^\ell \left( g_{kn \to i}^\ell \right)^2 \right) \tag{51}$$

and the following non-cavity ones:

$$\omega_{kn}^\ell = \sum_i m_{ki \to n}^\ell \, \hat{x}_{in \to k}^\ell \tag{52}$$

$$V_{kn}^\ell = \sum_i \left( \sigma_{ki \to n}^\ell \Delta_{in \to k}^\ell + \left( m_{ki \to n}^\ell \right)^2 \Delta_{in \to k}^\ell + \sigma_{ki \to n}^\ell \left( \hat{x}_{i'n \to k}^\ell \right)^2 \right) \tag{53}$$

we can express the first 2 cumulants of the upgoing messages as:

$$\hat{x}_{in \to k}^\ell = \partial_B \varphi^\ell(B_{in \to k}^\ell, A_{in \to k}^\ell, \omega_{in}^{\ell-1}, V_{in}^{\ell-1}) \tag{54}$$

$$\Delta_{in \to k}^\ell = \partial_B^2 \varphi^\ell(B_{in \to k}^\ell, A_{in \to k}^\ell, \omega_{in}^{\ell-1}, V_{in}^{\ell-1}) \tag{55}$$

**Wrapping it up**   Additional but straightforward considerations are required for the final input and output layers ($\ell = 0$ and $\ell = L$ respectively), since they do not receive messages from below and above respectively. In the end, thanks to independence assumptions and the central limit theorem that we used throughout the derivations, we arrive to a closed set of equations involving the means and the variances (or otherwise the corresponding natural parameters) of the messages. Within the same approximation assumption, we also replace the cavity quantities corresponding to variances with the non-cavity counterparts. Dividing the update equations in a *forward* and *backward* pass, and ordering them using time indexes in such a way that we have an efficient flow of information, we obtain the set of BP equations presented in the main text Eqs. (6-17) and in the Appendix Eqs. (60-71).

## A.3   BP EQUATIONS

We report here the end result of the derivation in last section, the complete set of BP equations also presented in the main text as Eqs. (6-17).

**Initialization** At $\tau = 0$:

$$B_{in \to k}^{\ell,0} = 0 \tag{56}$$

$$A_{in}^{\ell,0} = 0 \tag{57}$$

$$H_{ki \to n}^{\ell,0} = 0 \tag{58}$$

$$G_{ki}^{\ell,0} = 0 \tag{59}$$

**Forward Pass** At each $\tau = 1, \ldots, \tau_{max}$, for $\ell = 0, \ldots, L$:

$$\hat{x}_{in \to k}^{\ell,\tau} = \partial_B \varphi^\ell(B_{in \to k}^{\ell,\tau-1}, A_{in}^{\ell,\tau-1}, \omega_{in}^{\ell-1,\tau}, V_{in}^{\ell-1,\tau}) \tag{60}$$

$$\Delta_{in}^{\ell,\tau} = \partial_B^2 \varphi^\ell(B_{in}^{\ell,\tau-1}, A_{in}^{\ell,\tau-1}, \omega_{in}^{\ell-1,\tau}, V_{in}^{\ell-1,\tau}) \tag{61}$$

$$m_{ki \to n}^{\ell,\tau} = \partial_H \psi(H_{ki \to n}^{\ell,\tau-1}, G_{ki}^{\ell,\tau-1}, \theta_{ki}^\ell) \tag{62}$$

$$\sigma_{ki}^{\ell,\tau} = \partial_H^2 \psi(H_{ki}^{\ell,\tau-1}, G_{ki}^{\ell,\tau-1}, \theta_{ki}^\ell) \tag{63}$$

$$V_{kn}^{\ell,\tau} = \sum_i \left( \left(m_{ki}^{\ell,\tau}\right)^2 \Delta_{in}^{\ell,\tau} + \sigma_{ki}^{\ell,\tau-1} \left(\hat{x}_{i'n}^{\ell,\tau}\right)^2 + \sigma_{ki}^{\ell,\tau-1} \Delta_{in}^{\ell,\tau} \right) \tag{64}$$

$$\omega_{kn \to i}^{\ell,\tau} = \sum_{i' \neq i} m_{ki' \to n}^{\ell,\tau} \hat{x}_{i'n \to k}^{\ell,\tau} \tag{65}$$

In these equations for simplicity we abused the notation, in fact for the first layer $\hat{x}_n^{\ell=0,\tau}$ is fixed and given by the input $x_n$ while $\Delta_n^{\ell=0,\tau} = 0$ instead.

**Backward Pass** For $\tau = 1, \ldots, \tau_{max}$, for $\ell = L, \ldots, 0$ :

$$g_{kn \to i}^{\ell,\tau} = \partial_\omega \varphi^{\ell+1}(B_{kn}^{\ell+1,\tau}, A_{kn}^{\ell+1,\tau}, \omega_{kn \to i}^{\ell,\tau}, V_{kn}^{\ell,\tau}) \tag{66}$$

$$\Gamma_{kn}^{\ell,\tau} = -\partial_\omega^2 \varphi^{\ell+1}(B_{kn}^{\ell+1,\tau}, A_{kn}^{\ell+1,\tau}, \omega_{kn}^{\ell,\tau}, V_{kn}^{\ell,\tau}) \tag{67}$$

$$A_{in}^{\ell,\tau} = \sum_k \left( \left( \left(m_{ki}^{\ell,\tau}\right)^2 + \sigma_{ki}^{\ell,\tau} \right) \Gamma_{kn}^{\ell,\tau} - \sigma_{ki}^{\ell,\tau} \left(g_{kn}^{\ell,\tau}\right)^2 \right) \tag{68}$$

$$B_{in \to k}^{\ell,\tau} = \sum_{k' \neq k} m_{k'i \to n}^{\ell,\tau} g_{k'n \to i}^{\ell,\tau} \tag{69}$$

$$G_{ki}^{\ell,\tau} = \sum_n \left( \left( \left(\hat{x}_{in}^{\ell,\tau}\right)^2 + \Delta_{in}^{\ell,\tau} \right) \Gamma_{kn}^{\ell,\tau} - \Delta_{in}^{\ell,\tau} \left(g_{kn}^{\ell,\tau}\right)^2 \right) \tag{70}$$

$$H_{ki \to n}^{\ell,\tau} = \sum_{n' \neq n} \hat{x}_{in' \to k}^{\ell,\tau} g_{kn' \to i}^{\ell,\tau} \tag{71}$$

In these equations as well we abused the notation: calling $L$ the number of hidden neuron layers, when $\ell = L$ one should use $\varphi^{L+1}(y, \omega, V)$ from Eq. 21 instead of $\varphi^{L+1}(B, A, \omega, V)$.

## A.4 BPI EQUATIONS

The BP-Inspired algorithm (BPI) is obtained as an approximation of BP replacing some cavity quantities with their non-cavity counterparts. What we obtain is a generalization of the single layer algorithm of Baldassi et al. (2007).

**Forward pass.**

$$\hat{x}_{in}^{\ell,\tau} = \partial_B \varphi^\ell \left( B_{in}^{\ell,\tau-1}, A_{in}^{\ell,\tau-1}, \omega_{in}^{\ell-1,\tau}, V_{in}^{\ell-1,\tau} \right) \tag{72}$$

$$\Delta_{in}^{\ell,\tau} = \partial_B^2 \varphi^\ell \left( B_{in}^{\ell,\tau-1}, A_{in}^{\ell,\tau-1}, \omega_{in}^{\ell-1,\tau}, V_{in}^{\ell-1,\tau} \right) \tag{73}$$

$$m_{ki}^{\ell,\tau} = \partial_H \psi(H_{ki}^{\ell,\tau-1}, G_{ki}^{\ell,\tau-1}, \theta_{ki}^\ell) \tag{74}$$

$$\sigma_{ki}^{\ell,\tau} = \partial_H^2 \psi(H_{ki}^{\ell,\tau-1}, G_{ki}^{\ell,\tau-1}, \theta_{ki}^\ell) \tag{75}$$

$$V_{kn}^{\ell,\tau} = \sum_i \left( \left( m_{ki}^{\ell,\tau} \right)^2 \Delta_{in}^{\ell,\tau} + \sigma_{ki}^{\ell,\tau} (\hat{x}_{in}^{\ell,\tau})^2 + \sigma_{ki}^{\ell,\tau} \Delta_{in}^{\ell,\tau} \right) \tag{76}$$

$$\omega_{kn}^{\ell,\tau} = \sum_i m_{ki}^{\ell,\tau} \hat{x}_{in}^{\ell,\tau} \tag{77}$$

**Backward pass.**

$$g_{kn\to i}^{\ell,\tau} = \partial_\omega \varphi^{\ell+1} \left( B_{kn}^{\ell+1,\tau}, A_{kn}^{\ell+1,\tau}, \omega_{kn}^{\ell,\tau} - m_{ki}^{\ell,\tau} \hat{x}_{ai}^{\ell,\tau}, V_{kn}^{\ell,\tau} \right) \tag{78}$$

$$\Gamma_{kn}^{\ell,\tau} = -\partial_\omega^2 \varphi^{\ell+1} \left( B_{kn}^{\ell+1,\tau}, A_{kn}^{\ell+1,\tau}, \omega_{kn}^{\ell,\tau}, V_{kn}^{\ell,\tau} \right) \tag{79}$$

$$A_{in}^{\ell,\tau} = \sum_k \left( (m_{ki}^{\ell,\tau})^2 + \sigma_{ki}^{\ell,\tau} \right) \Gamma_{kn}^{\ell,\tau} - \sigma_{ki}^{\ell,\tau} \left( g_{kn}^{\ell,\tau} \right)^2 \tag{80}$$

$$B_{in}^{\ell,\tau} = \sum_k m_{ki}^{\ell,\tau} g_{kn\to i}^{\ell,\tau} \tag{81}$$

$$G_{ki}^{\ell,\tau} = \sum_n \left( (\hat{x}_{in}^{\ell,\tau})^2 + \Delta_{in}^{\ell,\tau} \right) \Gamma_{kn}^{\ell,\tau} - \Delta_{in}^{\ell,\tau} \left( g_{kn}^{\ell,\tau} \right)^2 \tag{82}$$

$$H_{ki}^{\ell,\tau} = \sum_n \hat{x}_{in}^{\ell,\tau} g_{kn\to i}^{\ell,\tau} \tag{83}$$

## A.5 MF EQUATIONS

The mean-field (MF) equations are obtained as a further simplification of BPI, using only non-cavity quantities. Although the simplification appears minimal at this point, we empirically observe a non-negligible discrepancy between the two algorithms in terms of generalization performance and computational time.

**Forward pass.**

$$\hat{x}_{in}^{\ell,\tau} = \partial_B \varphi^\ell \left( B_{in}^{\ell,\tau-1}, A_{in}^{\ell,\tau-1}, \omega_{in}^{\ell-1,\tau}, V_{in}^{\ell-1,\tau} \right) \tag{84}$$

$$\Delta_{in}^{\ell,\tau} = \partial_B^2 \varphi^\ell \left( B_{in}^{\ell,\tau-1}, A_{in}^{\ell,\tau-1}, \omega_{in}^{\ell-1,\tau}, V_{in}^{\ell-1,\tau} \right) \tag{85}$$

$$m_{ki}^{\ell,\tau} = \partial_H \psi(H_{ki}^{\ell,\tau-1}, G_{ki}^{\ell,\tau-1}, \theta_{ki}^\ell) \tag{86}$$

$$\sigma_{ki}^{\ell,\tau} = \partial_H^2 \psi(H_{ki}^{\ell,\tau-1}, G_{ki}^{\ell,\tau-1}, \theta_{ki}^\ell) \tag{87}$$

$$V_{kn}^{\ell,\tau} = \sum_i \left( \left( m_{ki}^{\ell,\tau} \right)^2 \Delta_{in}^{\ell,\tau} + \sigma_{ki}^{\ell,\tau} (\hat{x}_{in}^{\ell,\tau})^2 + \sigma_{ki}^{\ell,\tau} \Delta_{in}^{\ell,\tau} \right) \tag{88}$$

$$\omega_{kn}^{\ell,\tau} = \sum_i m_{ki}^{\ell,\tau} \hat{x}_{in}^{\ell,\tau} \tag{89}$$

**Backward pass.**

$$g_{kn}^{\ell,\tau} = \partial_\omega \varphi^{\ell+1}\left(B_{kn}^{\ell+1,\tau}, A_{kn}^{\ell+1,\tau}, \omega_{kn}^{\ell,\tau}, V_{kn}^{\ell,\tau}\right) \tag{90}$$

$$\Gamma_{kn}^{\ell,\tau} = -\partial_\omega^2 \varphi^{\ell+1}\left(B_{kn}^{\ell+1,\tau}, A_{kn}^{\ell+1,\tau}, \omega_{kn}^{\ell,\tau}, V_{kn}^{\ell,\tau}\right) \tag{91}$$

$$A_{in}^{\ell,\tau} = \sum_k \left((m_{ki}^{\ell,\tau})^2 + \sigma_{ki}^{\ell,\tau}\right)\Gamma_{kn}^{\ell,\tau} - \sigma_{ki}^{\ell,\tau}\left(g_{kn}^{\ell,\tau}\right)^2 \tag{92}$$

$$B_{in}^{\ell,\tau} = \sum_k m_{ki}^{\ell,\tau} g_{kn}^{\ell,\tau} \tag{93}$$

$$G_{ki}^{\ell,\tau} = \sum_n \left((\hat{x}_{in}^{\ell,\tau})^2 + \Delta_{in}^{\ell,\tau}\right)\Gamma_{kn}^{\ell,\tau} - \Delta_{in}^{\ell,\tau}\left(g_{kn}^{\ell,\tau}\right)^2 \tag{94}$$

$$H_{ki}^{\ell,\tau} = \sum_n \hat{x}_{in}^{\ell,\tau} g_{kn}^{\ell,\tau} \tag{95}$$

## A.6 DERIVATION OF THE AMP EQUATIONS

In order to obtain the AMP equations, we approximate cavity quantities with non-cavity ones in the BP equations Eqs. (60-71) using a first order expansion. We start with the mean activation:

$$
\begin{aligned}
\hat{x}_{in\to k}^{\ell,\tau} &= \partial_B \varphi^\ell(B_{in}^{\ell,\tau-1} - m_{ki\to n}^{\ell,\tau-1} g_{kn\to i}^{\ell,\tau-1}, A_{in}^{\ell,\tau-1}, \omega_{in}^{\ell-1,\tau}, V_{in}^{\ell-1,\tau}) \\
&\approx \partial_B \varphi^\ell(B_{in}^{\ell,\tau-1}, A_{in}^{\ell,\tau-1}, \omega_{in}^{\ell-1,\tau}, V_{in}^{\ell-1,\tau}) \\
&\quad - m_{ki\to n}^{\ell,\tau-1} g_{kn\to i}^{\ell,\tau-1} \partial_B^2 \varphi^\ell(B_{in}^{\ell,\tau-1}, A_{in}^{\ell,\tau-1}, \omega_{in}^{\ell-1,\tau}, V_{in}^{\ell-1,\tau}) \\
&\approx \hat{x}_{in}^{\ell,\tau} - m_{ki}^{\ell,\tau-1} g_{kn}^{\ell,\tau-1} \Delta_{in}^{\ell,\tau}
\end{aligned}
\tag{96}
$$

Analogously, for the weight's mean we have:

$$
\begin{aligned}
m_{ki\to n}^{\ell,\tau} &= \partial_H \psi(H_{ki}^{\ell,\tau-1} - \hat{x}_{in\to k}^{\ell,\tau-1} g_{kn\to i}^{\ell,\tau-1}, G_{ki}^{\ell,\tau-1}, \theta_{ki}^\ell) \\
&\approx \partial_H \psi(H_{ki}^{\ell,\tau-1}, G_{ki}^{\ell,\tau-1}, \theta_{ki}^\ell) - \hat{x}_{in\to k}^{\ell,\tau-1} g_{kn\to i}^{\ell,\tau-1} \partial_H^2 \psi(H_{ki}^{\ell,\tau-1}, G_{ki}^{\ell,\tau-1}, \theta_{ki}^\ell) \\
&\approx m_{ki}^{\ell,\tau} - \hat{x}_{in}^{\ell,\tau-1} g_{kn}^{\ell,\tau-1} \sigma_{ki}^{\ell,\tau}.
\end{aligned}
\tag{97}
$$

This brings us to:

$$
\begin{aligned}
\omega_{kn}^{\ell,\tau} &= \sum_i m_{ki\to n}^{\ell,\tau} \hat{x}_{in\to k}^{\ell,\tau} \\
&\approx \sum_i m_{ki}^{\ell,\tau} \hat{x}_{in}^{\ell,\tau} - g_{kn}^{\ell,\tau-1} \sum_i \sigma_{ki}^{\ell,\tau} \hat{x}_{in}^{\ell,\tau} \hat{x}_{in}^{\ell,\tau-1} - g_{kn}^{\ell,\tau-1} \sum_i m_{ki}^{\ell,\tau} m_{ki}^{\ell,\tau-1} \Delta_{in}^{\ell,\tau} \\
&\quad + (g_{kn}^{\ell,\tau-1})^2 \sum_i \sigma_{ki}^{\ell,\tau} m_{ki}^{\ell,\tau-1} \hat{x}_{in}^{\ell,\tau-1} \Delta_{in}^{\ell,\tau}
\end{aligned}
\tag{98}
$$

Let us now apply the same procedure to the other set of cavity messages:

$$
\begin{aligned}
g_{kn\to i}^{\ell,\tau} &= \partial_\omega \varphi^{\ell+1}(B_{kn}^{\ell+1,\tau}, A_{kn}^{\ell+1,\tau}, \omega_{kn}^{\ell,\tau} - m_{ki\to n}^{\ell,\tau} \hat{x}_{in\to k}^{\ell,\tau}, V_{kn}^{\ell,\tau}) \\
&\approx \partial_\omega \varphi^{\ell+1}(B_{kn}^{\ell+1,\tau}, A_{kn}^{\ell+1,\tau}, \omega_{kn}^{\ell,\tau}, V_{kn}^{\ell,\tau}) \\
&\quad - m_{ki\to n}^{\ell,\tau} \hat{x}_{in\to k}^{\ell,\tau} \partial_\omega^2 \varphi^{\ell+1}(B_{kn}^{\ell+1,\tau}, A_{kn}^{\ell+1,\tau}, \omega_{kn}^{\ell,\tau}, V_{kn}^{\ell,\tau}) \\
&\approx g_{kn}^{\ell,\tau} + m_{ki}^{\ell,\tau} \hat{x}_{in}^{\ell,\tau} \Gamma_{kn}^{\ell,\tau}
\end{aligned}
\tag{99}
$$

$$B_{in}^{\ell,\tau} = \sum_k m_{ki \to n}^{\ell,\tau} \, g_{kn \to i}^{\ell,\tau}$$

$$\approx \sum_k m_{ki}^{\ell,\tau} \, g_{kn}^{\ell,\tau} - \hat{x}_{in} \sum_k \left(g_{kn}^{\ell,\tau}\right)^2 \sigma_{ki}^{\ell,\tau} + \hat{x}_{in}^{\ell,\tau} \sum_k (m_{ki}^{\ell,\tau})^2 \Gamma_{kn}^{\ell,\tau}$$

$$- (\hat{x}_{in}^{\ell,\tau})^2 \sum_k \sigma_{ki}^{\ell,\tau} m_{ki}^{\ell,\tau} g_{kn}^{\ell,\tau} \Gamma_{kn}^{\ell,\tau} \tag{100}$$

$$H_{ki}^{\ell,\tau} = \sum_n \hat{x}_{in \to k}^{\ell,\tau} \, g_{kn \to i}^{\ell,\tau}$$

$$\approx \sum_n \hat{x}_{in}^{\ell,\tau} \, g_{kn}^{\ell,\tau} + m_{ki}^{\ell,\tau} \sum_n \left(\hat{x}_{in}^{\ell,\tau}\right)^2 \Gamma_{kn}^{\ell,\tau} - m_{ki}^{\ell,\tau} \sum_n (g_{kn}^{\ell,\tau})^2 \Delta_{in}^{\ell,\tau}$$

$$- (m_{ki}^{\ell,\tau})^2 \sum_n g_{kn}^{\ell,\tau} \Gamma_{kn}^{\ell,\tau} \Delta_{in}^{\ell,\tau} \hat{x}_{in}^{\ell,\tau} \tag{101}$$

We are now able to write down the full AMP equations, that we present in the next section.

### A.7 AMP EQUATIONS

In summary, in the last section we derived the AMP algorithm as a closure of the BP messages passing over non-cavity quantities, relying on some statistical assumptions on messages and interactions. With respect to the MF message passing, we find some additional terms that go under the name of Onsager corrections. In-depth overviews of the AMP (also known as Thouless-Anderson-Palmer (TAP)) approach can be found in Refs. (Zdeborová & Krzakala, 2016; Mézard, 2017; Gabrié, 2020). The final form of the AMP equations for the multi-layer perceptron is given below.

**Initialization**   At $\tau = 0$:

$$B_{in}^{\ell,0} = 0 \tag{102}$$

$$A_{in}^{\ell,0} = 0 \tag{103}$$

$$H_{ki}^{\ell,0} = 0 \text{ or some values} \tag{104}$$

$$G_{ki}^{\ell,0} = 0 \text{ or some values} \tag{105}$$

$$g_{kn}^{\ell,0} = 0 \tag{106}$$

**Forward Pass**   At each $\tau = 1, \ldots, \tau_{max}$, for $\ell = 0, \ldots, L$:

$$\hat{x}_{in}^{\ell,\tau} = \partial_B \varphi^\ell(B_{in}^{\ell,\tau-1}, A_{in}^{\ell,\tau-1}, \omega_{in}^{\ell-1,\tau}, V_{in}^{\ell-1,\tau}) \tag{107}$$

$$\Delta_{in}^{\ell,\tau} = \partial_B^2 \varphi^\ell(B_{in}^{\ell,\tau-1}, A_{in}^{\ell,\tau-1}, \omega_{in}^{\ell-1,\tau}, V_{in}^{\ell-1,\tau}) \tag{108}$$

$$m_{ki}^{\ell,\tau} = \partial_H \psi(H_{ki}^{\ell,\tau-1}, G_{ki}^{\ell,\tau-1}, \theta_{ki}^\ell) \tag{109}$$

$$\sigma_{ki}^{\ell,\tau} = \partial_H^2 \psi(H_{ki}^{\ell,\tau-1}, G_{ki}^{\ell,\tau-1}, \theta_{ki}^\ell) \tag{110}$$

$$V_{kn}^{\ell,\tau} = \sum_i \left( \left(m_{ki}^{\ell,\tau}\right)^2 \Delta_{in}^{\ell,\tau} + \sigma_{ki}^{\ell,\tau} \left(\hat{x}_{i'n}^{\ell,\tau}\right)^2 + \sigma_{ki}^{\ell,\tau} \Delta_{in}^{\ell,\tau} \right) \tag{111}$$

$$\omega_{kn}^{\ell,\tau} = \sum_i m_{ki}^{\ell,\tau} \hat{x}_{in}^{\ell,\tau} - g_{kn}^{\ell,\tau-1} \sum_i \sigma_{ki}^{\ell,\tau} \hat{x}_{in}^{\ell,\tau} \hat{x}_{in}^{\ell,\tau-1} - g_{kn}^{\ell,\tau-1} \sum_i m_{ki}^{\ell,\tau} m_{ki}^{\ell,\tau-1} \Delta_{in}^{\ell,\tau}$$

$$+ (g_{kn}^{\ell,\tau-1})^2 \sum_i \sigma_{ki}^{\ell,\tau} m_{ki}^{\ell,\tau-1} \hat{x}_{in}^{\ell,\tau-1} \Delta_{in}^{\ell,\tau} \tag{112}$$

**Backward Pass**

$$g_{kn}^{\ell,\tau} = \partial_\omega \varphi^{\ell+1}(B_{kn}^{\ell+1,\tau}, A_{kn}^{\ell+1,\tau}, \omega_{kn \to i}^{\ell,\tau}, V_{kn}^{\ell,\tau}) \tag{113}$$

$$\Gamma_{kn}^{\ell,\tau} = -\partial_\omega^2 \varphi^{\ell+1}(B_{kn}^{\ell+1,\tau}, A_{kn}^{\ell+1,\tau}, \omega_{kn}^{\ell,\tau}, V_{kn}^{\ell,\tau}) \tag{114}$$

$$A_{in}^{\ell,\tau} = \sum_k \left( \left( \left( m_{ki}^{\ell,\tau} \right)^2 + \sigma_{ki}^{\ell,\tau} \right) \Gamma_{kn}^{\ell,\tau} - \sigma_{ki}^{\ell,\tau} \left( g_{kn}^{\ell,\tau} \right)^2 \right) \tag{115}$$

$$B_{in}^{\ell,\tau} = \sum_k m_{ki}^{\ell,\tau} \, g_{kn}^{\ell,\tau} - \hat{x}_{in} \sum_k \left( g_{kn}^{\ell,\tau} \right)^2 \sigma_{ki}^{\ell,\tau} + \hat{x}_{in}^{\ell,\tau} \sum_k (m_{ki}^{\ell,\tau})^2 \Gamma_{kn}^{\ell,\tau}$$
$$- (\hat{x}_{in}^{\ell,\tau})^2 \sum_k \sigma_{ki}^{\ell,\tau} m_{ki}^{\ell,\tau} g_{kn}^{\ell,\tau} \Gamma_{kn}^{\ell,\tau} \tag{116}$$

$$G_{ki}^{\ell,\tau} = \sum_n \left( \left( \left( \hat{x}_{in}^{\ell,\tau} \right)^2 + \Delta_{in}^{\ell,\tau} \right) \Gamma_{kn}^{\ell,\tau} - \Delta_{in}^{\ell,\tau} \left( g_{kn}^{\ell,\tau} \right)^2 \right) \tag{117}$$

$$H_{ki}^{\ell,\tau} = \sum_n \hat{x}_{in}^{\ell,\tau} \, g_{kn}^{\ell,\tau} + m_{ki}^{\ell,\tau} \sum_n \left( \hat{x}_{in}^{\ell,\tau} \right)^2 \Gamma_{kn}^{\ell,\tau} - m_{ki}^{\ell,\tau} \sum_n (g_{kn}^{\ell,\tau})^2 \Delta_{in}^{\ell,\tau}$$
$$- (m_{ki}^{\ell,\tau})^2 \sum_n g_{kn}^{\ell,\tau} \Gamma_{kn}^{\ell,\tau} \Delta_{in}^{\ell,\tau} \hat{x}_{in}^{\ell,\tau} \tag{118}$$

## A.8 ACTIVATION FUNCTIONS

### A.8.1 SIGN

In most of our experiments we use sign activations in each layer. With this choice, the neuron's free energy 19 takes the form

$$\varphi(B, A, \omega, V) = \log \left( \frac{1}{2} \sum_{x \in \{-1,+1\}} e^{Bx} \mathcal{H} \left( -\frac{x\omega}{\sqrt{V}} \right) \right) + \frac{1}{2} \log(2\pi V), \tag{119}$$

where

$$\mathcal{H} = \frac{1}{2} \operatorname{erfc} \left( \frac{x}{\sqrt{2}} \right). \tag{120}$$

Notice that for sign activations the messages $A$ can be dropped.

### A.8.2 RELU

For $ReLU(x) = \max(0, x)$ activations the free energy 19 becomes

$$\varphi(B, A, \omega, V) = \int \mathrm{d}x \mathrm{d}z \, e^{-\frac{1}{2}Ax^2 + Bx} \, \delta(x - \max(0, z)) \, e^{-\frac{(\omega - z)^2}{2V}} \tag{121}$$

$$= \log \left( H \left( \frac{\omega}{\sqrt{V}} \right) + \frac{\mathcal{N}(\omega; B/A, V + \frac{1}{A})}{A \mathcal{N}(B; 0, A)} H \left( -\frac{BV + \omega}{\sqrt{V + AV^2}} \right) \right) + \frac{1}{2} \log(2\pi V), \tag{122}$$

where

$$\mathcal{N}(x; \mu, \Sigma) = \frac{1}{\sqrt{2\pi\Sigma}} e^{-\frac{(x-\mu)^2}{2\Sigma}}. \tag{123}$$

## A.9 THE ARGMAX LAYER

In order to perform multi-class classification, we have to perform an argmax operation on the last layer of the neural network. Call $z_k$, for $k = 1, \ldots, K$, the Gaussian random variables output of the last layer of the network in correspondence of some input $x$. Assuming the correct label is class $k^*$, the effective partition function $Z_{k^*}$ corresponding to the output constraint reads:

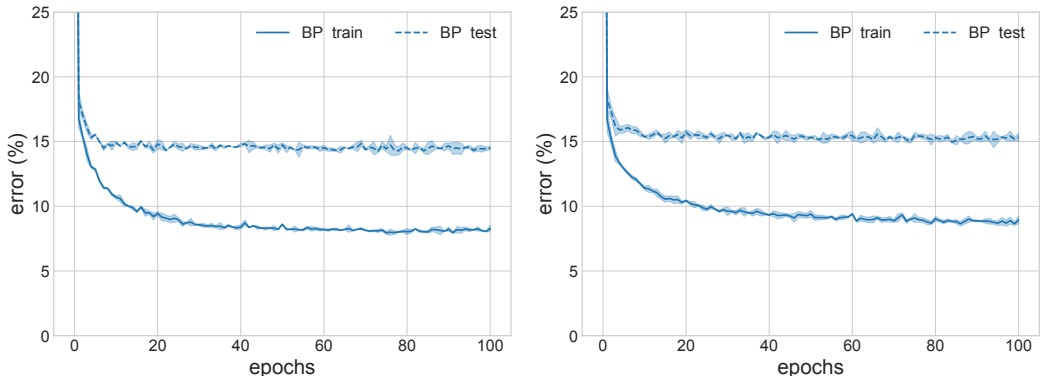

Figure 4: MLP with 2 hidden layers with 101 hidden units each, batch-size 128 on the Fashion-MNIST dataset. In the first two layers we use the BP equations, while in the last layer the ArgMax ones. (Left) ArgMax layer first version; (Right) ArgMax layer second version. Even if it is possible to reach similar accuracies with the two versions, we decide to use the first one as it is simpler to use.

$$Z_{k^*} = \int \prod_k dz_k \, \mathcal{N}(z_k; \omega_k, V_k) \prod_{k \neq k^*} \Theta(z_{k^*} - z_k) \tag{124}$$

$$= \int dz_{k^*} \, \mathcal{N}(z_{k^*}; \omega_{k^*}, V_{k^*}) \prod_{k \neq k^*} \mathcal{H}\left(-\frac{z_{k^*} - \omega_k}{\sqrt{V_k}}\right) \tag{125}$$

Here $\Theta(x)$ is the Heaviside indicator function and we used the definition of $\mathcal{H}$ from Eq. 120. The integral on the last line cannot be expressed analytically, therefore we have to resort to approximations.

### A.9.1 APPROACH 1: JENSEN INEQUALITY

Using the Jensen inequality we obtain:

$$\phi_{k^*} = \log Z_{k^*} = \log \mathbb{E}_{z \sim \mathcal{N}(\omega_{k^*}, V_{k^*})} \prod_{k \neq k^*} \mathcal{H}\left(-\frac{z - \omega_k}{\sqrt{V_k}}\right) \tag{126}$$

$$\geq \sum_{k \neq k^*} \mathbb{E}_{z \sim \mathcal{N}(\omega_{k^*}, V_{k^*})} \log \mathcal{H}\left(-\frac{z - \omega_k}{\sqrt{V_k}}\right) \tag{127}$$

Reparameterizing the expectation we have:

$$\tilde{\phi}_{k^*} = \sum_{k \neq k^*} \mathbb{E}_{\epsilon \sim \mathcal{N}(0,1)} \log \mathcal{H}\left(-\frac{\omega_{k^*} + \epsilon\sqrt{V_{k^*}} - \omega_k}{\sqrt{V_k}}\right) \tag{128}$$

The derivative $\partial_{\omega_k} \tilde{\phi}_{k^*}$ and $\partial^2_{\omega_k} \tilde{\phi}_{k^*}$ that we need can then be estimated by sampling (once) $\epsilon$:

$$\partial_{\omega_k} \tilde{\phi}_{k^*} = \begin{cases} -\frac{1}{\sqrt{V_k}} \mathbb{E}_{\epsilon \sim \mathcal{N}(0,1)} \mathcal{K}\left(-\frac{\omega_{k^*} + \epsilon\sqrt{V_{k^*}} - \omega_k}{\sqrt{V_k}}\right) & k \neq k^* \\ \sum_{k' \neq k^*} \frac{1}{\sqrt{V_{k'}}} \mathbb{E}_{\epsilon \sim \mathcal{N}(0,1)} \mathcal{K}\left(-\frac{\omega_{k^*} + \epsilon\sqrt{V_{k^*}} - \omega_{k'}}{\sqrt{V_{k'}}}\right) & k = k^* \end{cases} \tag{129}$$

where we have defined:

$$\mathcal{K}(x) = \frac{\mathcal{N}(x)}{\mathcal{H}(x)} = \frac{\sqrt{2/\pi}}{\text{erfcx}(x/2)} \tag{130}$$

### A.9.2 APPROACH 2: JENSEN AGAIN

A further simplification is obtained by applying Jensen inequality again to 128 but in the opposite direction, therefore we renounce to having a bound and look only for an approximation. We have the new effective free energy:

$$\tilde{\phi}_{k^*} = \sum_{k \neq k^*} \log \mathbb{E}_{\epsilon \sim \mathcal{N}(0,1)} \mathcal{H} \left( -\frac{\omega_{k^*} + \epsilon \sqrt{V_{k^*}} - \omega_k}{\sqrt{V_k}} \right) \tag{131}$$

$$= \sum_{k \neq k^*} \log \mathcal{H} \left( -\frac{\omega_{k^*} - \omega_k}{\sqrt{V_k + V_{k^*}}} \right) \tag{132}$$

This gives, for $k \neq k^*$:

$$\partial_{\omega_k} \tilde{\phi}_{k^*} = \begin{cases} -\frac{1}{\sqrt{V_k + V_{k^*}}} \mathcal{K} \left( -\frac{\omega_{k^*} - \omega_k}{\sqrt{V_k + V_{k^*}}} \right) & k \neq k^* \\ \sum_{k' \neq k^*} \frac{1}{\sqrt{V_{k'} + V_{k^*}}} \mathcal{K} \left( -\frac{\omega_{k^*} - \omega_{k'}}{\sqrt{V_{k'} + V_{k^*}}} \right) & k = k^* \end{cases} \tag{133}$$

Notice that $\partial_{\omega_{k^*}} \tilde{\phi}_{k^*} = -\sum_{k \neq k^*} \partial_{\omega_k} \tilde{\phi}_{k^*}$. In last formulas we used the definition of $\mathcal{K}$ in Eq. 130.

We show in Fig. 4 the negligible difference between the two ArgMax versions when using BP on the layers before the last one (which performs only the ArgMax).

## B EXPERIMENTAL DETAILS

### B.1 HYPER-PARAMETERS OF THE BP-BASED SCHEME

We include here a complete list of the hyper-parameters present in the BP-based algorithms. Notice that, like in the SGD type of algorithms, many of them can be fixed or it is possible to find a prescription for their value that works in most cases. However, we expect future research to find even more effective values of the hyper-parameters, in the same way it has been done for SGD. These hyper-parameters are: the mini-batch size $bs$; the parameter $\rho$ (that has to be tuned similarly to the learning rate in SGD); the damping parameter $\alpha$ (that performs a running smoothing on the BP fields along the dynamics by adding a fraction of the field at the previous iteration, see Eqs. (134, 135)); the initialization coefficient $\epsilon$ that we use to to sample the parameters of our prior distribution $q_\theta(\mathcal{W})$ according to $\theta_{ki}^{\ell,t=0} \sim \epsilon \mathcal{N}(0,1)$. Different choices of $\epsilon$ correspond to different initial distribution of the weights' magnetization $m_{ki}^\ell = \tanh(\theta_{ki}^\ell)$, as is shown in Fig. 5); the number of internal steps of reinforcement $\tau_{\max}$ and the associated intensity of the internal reinforcement $r$. The performances of the BP-based algorithms are robust in a reasonable range of these hyper-parameters. A more principled choice of a good initialization condition could be made by adapting the technique from Stamatescu et al. (2020).

Notice that among these parameters, the BP dynamics at each layer is mostly sensitive to $\rho$ and $\alpha$, so that in general we consider them layer-dependent. See Sec. B.8 for details on the effect of these parameters on the learning dynamics and on layer polarization (i.e. how the BP dynamics tends to bias the weights towards a single point-wise configuration with high probability). Unless otherwise stated we fix some of the hyper-parameters, in particular: $bs = 128$ (results are consistent with other values of the batch-size, from $bs = 1$ up to $bs = 1024$ in our experiments), $\epsilon = 1.0$, $\tau_{\max} = 1$, $r = 0$.

### B.2 DAMPING SCHEME FOR THE MESSAGE PASSING

We use a damping parameter $\alpha \in (0,1)$ to stabilize the training, changing the updated rule for the weights' means as follows

$$\tilde{m}_{ki}^{\ell,\tau} = \partial_H \psi(H_{ki}^{\ell,\tau-1}, G_{ki}^{\ell,\tau-1}, \theta_{ki}^\ell) \tag{134}$$

$$m_{ki}^{\ell,\tau} = \alpha \, m_{ki}^{\ell,\tau-1} + (1 - \alpha) \, \tilde{m}_{ki}^{\ell,\tau} \tag{135}$$

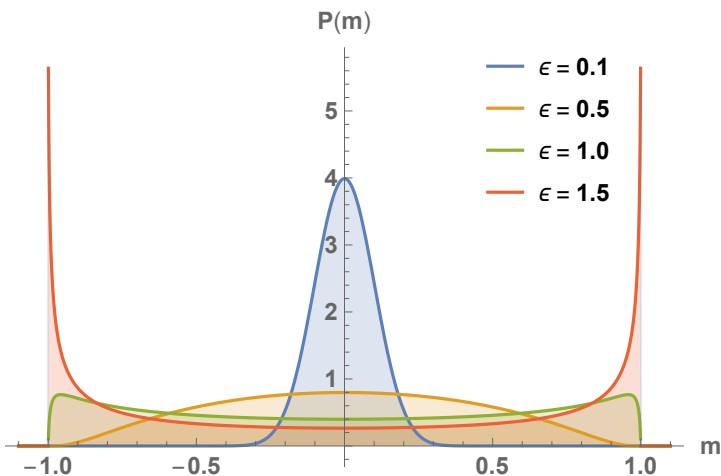

Figure 5: Initial distribution of the magnetizations varying the parameter $\epsilon$. The initial distribution is more concentrated around $\pm 1$ as $\epsilon$ increases (i.e. it is more bimodal and the initial configuration is more polarized).

### B.3 ARCHITECTURES

In the experiments in which we vary the architecture (see Sec. 4.1), all simulations of the BP-based algorithms use a number of internal reinforcement iterations $\tau_{\max} = 1$. Learning is performed on the totality of the training dataset, the batch-size is $bs = 128$, the initialization coefficient is $\epsilon = 1.0$.

For all architectures and all BP approximations, we use $\alpha = 0.8$ for each layer, apart for the 501-501-501 MLP in which we use $\alpha = (0.1, 0.1, 0.1, 0.9)$. Concerning the parameter $\rho$, we use $\rho = 0.9$ on the last layer for all architectures and BP approximations. On the other layers we use: for the 101-101 and the 501-501 MLPs, $\rho = 1.0001$ for all BP approximations; for the 101-101-101 MLP, $\rho = 1.0$ for BP and AMP while $\rho = 1.001$ for MF; for the 501-501-501 MLP $\rho = 1.0001$ for all BP approximations. For the BinaryNet simulations, the learning rate is $lr = 10.0$ for all MLP architectures, giving the better performance among the learning rates we have tested, $lr = 100, 10, 1, 0.1, 0.001$.

We notice that while we need some tuning of the hyper-parameters to reach the performances of BinaryNet, it is possible to fix them across datasets and architectures (e.g. $\rho = 1$ and $\alpha = 0.8$ on each layer) without in general losing more than $20\%$ (relative) of the generalization performances, demonstrating that the BP-based algorithms are effective for learning also with minimal hyper-parameter tuning.

The experiments on the Bayesian error are performed on a MLP with 2 hidden layers of 101 units on the MNIST dataset (binary classification). Learning is performed on the totality of the training dataset, the batch-size is $bs = 128$, the initialization coefficient is $\epsilon = 1.0$. In order to find the pointwise configurations we use $\alpha = 0.8$ on each layer and $\rho = (1.0001, 1.0001, 0.9)$, while to find the Bayesian ones we use $\alpha = 0.8$ on each layer and $\rho = (0.9999, 0.9999, 0.9)$ (these value prevent an excessive polarization of the network towards a particular pointwise configurations).

For the continual learning task (see Sec. 4.5) we fixed $\rho = 1$ and $\alpha = 0.8$ on each layer as we empirically observed that polarizing the last layer helps mitigating the forgetting while leaving the single-task performances almost unchanged.

In Fig. 6 we report training curves on architectures different from the ones reported in the main paper.

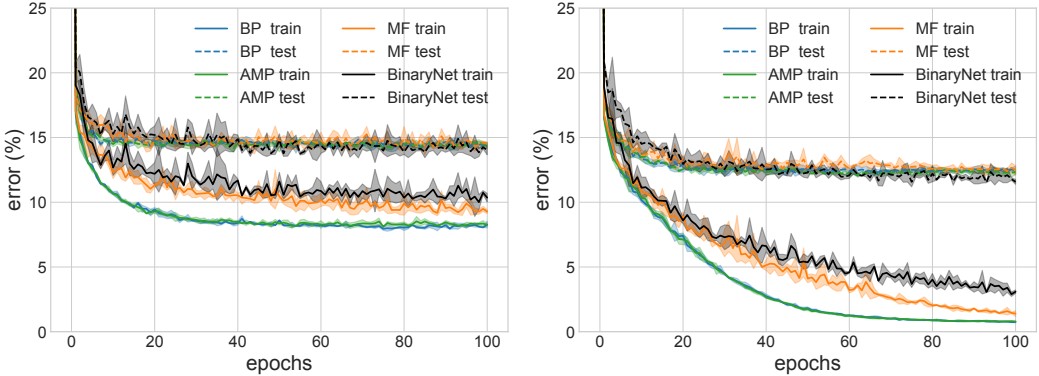

Figure 6: Training curves of message passing algorithms compared with BinaryNet on the Fashion-MNIST dataset (multi-class classification) with a binary MLP with 3 hidden layers of 501 units. (Right) The batch-size is 128 and curves are averaged over 5 realizations of the initial conditions

## B.4 VARYING THE DATASET

When varying the dataset (see Sec. 4.3), all simulation of the BP-based algorithms use a number of internal reinforcement iterations $\tau_{\max} = 1$. Learning is performed on the totality of the training dataset, the batch-size is $bs = 128$, the initialization coefficient is $\epsilon = 1.0$. For all datasets (MNIST (2 classes), FashionMNIST (2 classes), CIFAR-10 (2 classes), MNIST, FashionMNIST, CIFAR-10) and all algorithms (BP, AMP, MF) we use $\rho = (1.0001, 1.0001, 0.9)$ and $\alpha = 0.8$ for each layer. Using in the first layers values of $\rho = 1 + \epsilon$ with $\epsilon \geq 0$ and sufficiently small typically leads to good results.

For the BinaryNet simulations, the learning rate is $lr = 10.0$ (both for binary classification and multi-class classification), giving the better performance among the learning rates we have tested, $lr = 100, 10, 1, 0.1, 0.001$. In Tab. 2 we report the final train errors obtained on the different datasets.

| Dataset | BinaryNet | BP | AMP | MF |
|---|---|---|---|---|
| **MNIST (2 classes)** | $0.05 \pm 0.05$ | $0.0 \pm 0.0$ | $0.0 \pm 0.0$ | $0.0 \pm 0.0$ |
| **FashionMNIST (2 classes)** | $0.3 \pm 0.1$ | $0.06 \pm 0.01$ | $0.06 \pm 0.01$ | $0.09 \pm 0.01$ |
| **CIFAR10 (2 classes)** | $1.2 \pm 0.5$ | $0.37 \pm 0.01$ | $0.4 \pm 0.1$ | $0.9 \pm 0.2$ |
| **MNIST** | $0.09 \pm 0.01$ | $0.12 \pm 0.01$ | $0.12 \pm 0.01$ | $0.03 \pm 0.01$ |
| **FashionMNIST** | $4.0 \pm 0.5$ | $3.4 \pm 0.1$ | $3.7 \pm 0.1$ | $2.5 \pm 0.2$ |
| **CIFAR10** | $13.0 \pm 0.9$ | $4.7 \pm 0.1$ | $4.7 \pm 0.2$ | $9.2 \pm 0.5$ |

Table 2: Train error (%) on Fashion-MNIST of a multilayer perceptron with 2 hidden layers of 501 units each for BinaryNet (baseline), BP, AMP and MF. All algorithms are trained with batch-size 128 and for 100 epochs. Mean and standard deviations are calculated over 5 random initializations.

## B.5 LOCAL ENERGY

We adapt the notion of flatness used in (Jiang et al., 2020; Pittorino et al., 2021), that we call local energy, to configurations with binary weights. Given a weight configuration $\boldsymbol{w} \in \{\pm 1\}^N$, we define the local energy $\delta E_{\text{train}}(\boldsymbol{w}, p)$ as the average difference in training error $E_{\text{train}}(\boldsymbol{w})$ when perturbing $\boldsymbol{w}$ by flipping a random fraction $p$ of its elements:

$$\delta E_{\text{train}}(\boldsymbol{w}, p) = \mathbb{E}_{\boldsymbol{z}} \, E_{\text{train}}(\boldsymbol{w} \odot \boldsymbol{z}) - E_{\text{train}}(\boldsymbol{w}), \tag{136}$$

where $\odot$ denotes the Hadamard (element-wise) product and the expectation is over i.i.d. entries for $\boldsymbol{z}$ equal to $-1$ with probability $p$ and to $+1$ with probability $1 - p$. We report the resulting local energy profiles (in a range $[0, p_{\max}]$) in Fig. 7 right panel for BP and BinaryNet. The relative error grows

slowly when perturbing the trained configurations (notice the convexity of the curves). This shows that both BP-based and SGD-based algorithms find configurations that lie in relatively flat minima in the energy landscape. The same qualitative phenomenon holds for different architectures and datasets.

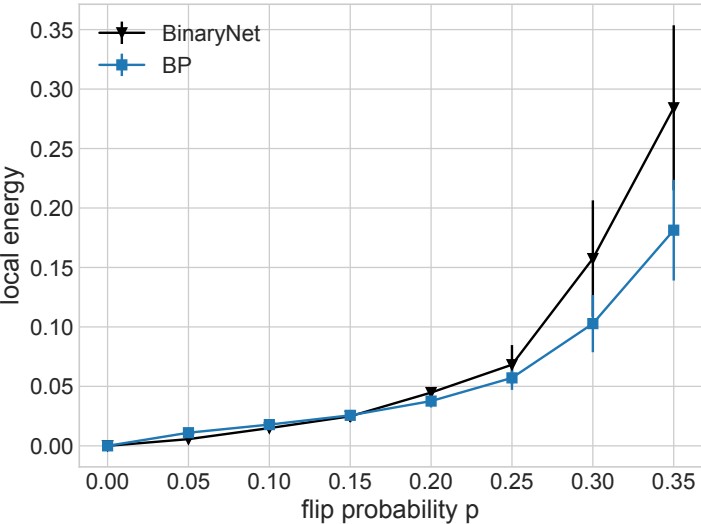

Figure 7: Local energy curve of the point-wise configuration found by the BP algorithm compared with BinaryNet on a MLP with 2 hidden layers of 101 units on the 2-class MNIST dataset.

### B.6 SGD IMPLEMENTATION (BINARYNET)

We compare the BP-based algorithms with SGD training for neural networks with binary weights and activations as introduced in BinaryNet (Hubara et al., 2016). This procedure consists in keeping a continuous version of the parameters $w$ which is updated with the SGD rule, with the gradient calculated on the binarized configuration $w_b = \text{sign}(w)$. At inference time the forward pass is calculated with the parameters $w_b$. The backward pass with binary activations is performed with the so called *straight-through estimator*.

Our implementation presents some differences with respect to the original proposal of the algorithm in (Hubara et al., 2016), in order to keep the comparison as fair as possible with the BP-based algorithms, in particular for what concerns the number of parameters. We do not use biases nor batch normalization layers, therefore in order to keep the pre-activations of each hidden layer normalized we rescale them by $\frac{1}{\sqrt{N}}$ where $N$ is the size of the previous layer (or the input size in the case of the pre-activations afferent to the first hidden layer). The standard SGD update rule is applied (instead of Adam), and we use the binary cross-entropy loss. Clipping of the continuous configuration $w$ in $[-1, 1]$ is applied. We use Xavier initialization (Glorot & Bengio, 2010) for the continuous weights. In Fig.2 of the main paper, we apply the Adam optimization rule, noticing that it performs slightly better in train and test generalization performance compared to the pure SGD one.

### B.7 EBP IMPLEMENTATION

Expectation Back Propagation (EBP) Soudry et al. (2014b) is parameter-free Bayesian algorithm that uses a mean-field (MF) approximation (fully factorized form for the posterior) in an online environment to estimate the Bayesian posterior distribution after the arrival of a new data point. The main differences between EBP and our approach relies in the approximation for the posterior distribution. Moreover we explicitly base the estimation of the marginals on the local high entropy structure. The fact that EBP works has no clear explanation: certainly it cannot be that the MF assumption holds for multi-layer neural networks. Still, it's certainly very interesting that it works. We argue that it might work precisely by virtue of the existence of high local entropy minima and

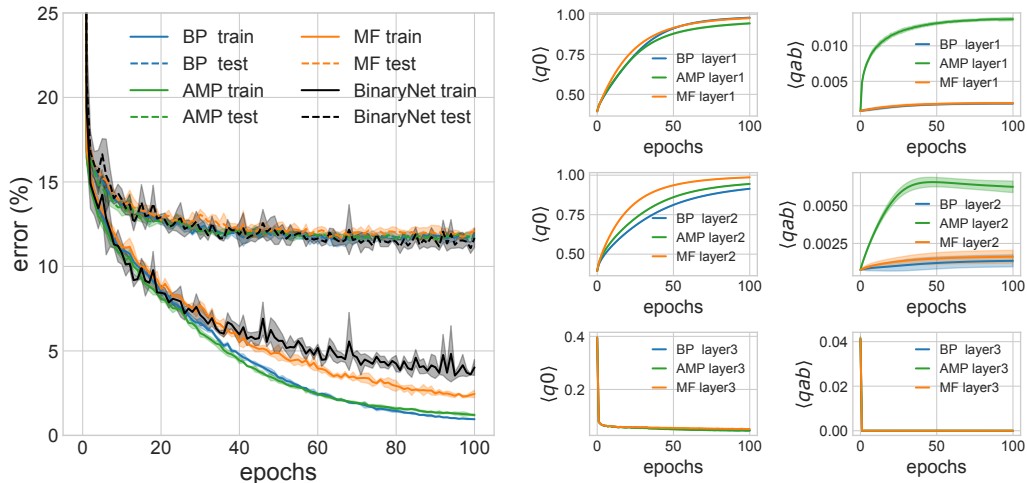

Figure 8: (Right panels) Polarizations $\langle q_0 \rangle$ and overlaps $\langle q_{ab} \rangle$ on each layer of a MLP with 2 hidden layers of 501 units on the Fashion-MNIST dataset (multi-class), the batch-size is $bs = 128$. (Right) Corresponding train and test error curves.

expect it to give similar performance to the MF case of our algorithm. The online iteration could in fact be seen as way of implementing a reinforcement.

We implemented the EBP code along the lines of the original matlab implementation (https://github.com/ExpectationBackpropagation/EBP_Matlab_Code). In order to perform a fair comparison we removed the biases both in the binary and continuous weights versions. It is worth noticing that we faced numerical issues in training with a moderate to big batchsize All the experiments were consequently limited to a batchsize of 10 patterns

### B.8   UNIT POLARIZATION AND OVERLAPS

We define the self-overlap or polarization of a given unit $a$ as $q_0^a = \frac{1}{N} \sum_i (w_i^a)^2$, where $N$ is the number of parameters of the unit and $\{w_i^a\}_{i=1}^N$ its weights. It quantifies how much the unit is polarized towards a unique point-wise binary configuration ($q_0^a = 1$ corresponding to high confidence in a given configurations while $q_0^a = 0$ to low). The overlap between two units $a$ and $b$ (considered in the same layer) is $q_{ab} = \frac{1}{N} \sum w_i^a w_i^b$. The number of parameters $N$ is the same for units belonging to the same fully connected layer. We denote by $\langle q_0 \rangle = \frac{1}{N_{out}} \sum_{a=1}^{N_{out}} q_0^a$ and $\langle q_{ab} \rangle = \frac{1}{N_{out}} \sum_{a<b}^{N_{out}} q_{ab}$ the mean polarization and mean overlap in a given layer (where $N_{out}$ is the number of units in the layer).

The parameters $\rho$ and $\alpha$ govern the dynamical evolution of the polarization of each layer during training. A value $\rho \gtrsim 1$ has the effect to progressively increase the units polarization during training, while $\rho < 1$ disfavours it. The damping $\alpha$ which takes values in $[0, 1]$ has the effect to slow the dynamics by a smoothing process (the intensity of which depends on the value of $\alpha$), generically favoring convergence. Given the nature of the updates in Algorithm 1, each layer presents its own dynamics given the values of $\rho_\ell$ and $\alpha \ell$ at layer $\ell$, that in general can differ from each other.

We find that it is is beneficial to control the polarization layer-per-layer, see Fig. 8 for the corresponding typical behavior of the mean polarization and the mean overlaps during training. Empirically, we have found that (as we could expect) when training is successful the layers polarize progressively towards $q_0 = 1$, i.e. towards a precise point-wise solution, while the overlaps between units in each hidden layer are such that $q_{ab} \ll 1$ (indicating low symmetry between intra-layer units, as expected for a non-redundant solution). To this aim, in most cases $\alpha \ell$ can be the same for each layer, while tuning $\rho_\ell$ for each layer allows to find better generalization performances in some cases (but is not strictly necessary for learning).

In particular, it is possible to use the same value $\rho_\ell$ for each layer before the last one ($\ell < L$ where $L$ is the number of layers in the network), while we have found that the last layer tends to

polarize immediately during the dynamics (probably due to its proximity to the output constraints). Empirically, it is usually beneficial for learning that this layer does not or only slightly polarize, i.e. $\langle q_0 \rangle \ll 1$ (this can be achieved by imposing $\rho_L < 1$). Learning is anyway possible even when the last layer polarizes towards $\langle q_0 \rangle = 1$ along the dynamics, i.e. by choosing $\rho_L$ sufficiently large.

As a simple general prescription in most experiments we can fix $\alpha = 0.8$ and $\rho_L = 0.9$, therefore leaving $\rho_{\ell < L}$ as the only hyper-parameter to be tuned, akin to the learning rate in SGD. Its value has to be very close to 1.0 (a value smaller than 1.0 tends to depolarize the layers, without focusing on a particular point-wise binary configuration, while a value greater than 1.0 tends to lead to numerical instabilities and parameters' divergence).

### B.9 COMPUTATIONAL PERFORMANCE: VARYING BATCH-SIZE

In order to compare the time performances of the BP-based algorithms with our implementation of BinaryNet, we report in Fig. 9 the time in seconds taken by a single epoch of each algorithm in function of the batch-size, on a MLP of 2 layers of 501 units on Fashion-MNIST. We test both algorithms on a NVIDIA GeForce RTX 2080 Ti GPU. Multi-class and binary classification present a very similar time scaling with the batch-size, in both cases comparable with BinaryNet. Let us also notice that BP-based algorithms are able to reach generalization performances comparable to BinaryNet for all the values of the batch-size reported in this section.

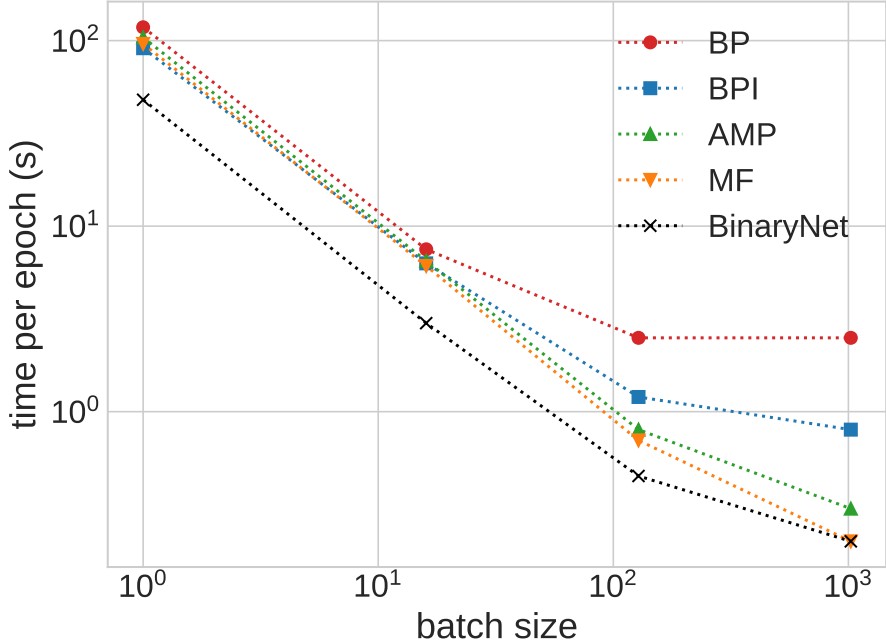

Figure 9: Algorithms time scaling with the batch-size on a MLP with 2 hidden layers of 501 hidden units each on the Fashion-MNIST dataset (multi-class classification). The reported time (in seconds) refers to one epoch for each algorithm.

