# OpenReview forum: "Deep learning via message passing algorithms based on belief propagation"
_ICLR.cc/2022/Conference — ICLR 2022 Submitted_

### Official Review · Reviewer_SnTy · 2021-10-27

**Correctness:** 4
**Technical Novelty And Significance:** 3
**Empirical Novelty And Significance:** 3
**Recommendation:** 8
**Confidence:** 3

**Main Review:**

Correctness:
No correctness concerns.

Technical novelty and significance:
One of the ideas of the paper---to use belief propagation to estimate the marginals of the weights of a neural network (NN) in order to make probabilistic predictions---is a novel paradigm allowing to attach confidence to NN predictions. This is a research direction with a lot of potential which can lead to better NN performance in many tasks.
However, the paper's theoretical contributions are somewhat limited as they seem to be very similar to the contributions of (Baldassi et al., 2016). Although the authors claim that the novelty of their approach is that their results apply to mini-batch training and deep neural networks, it is unclear how the results derived for deep neural networks differ from those derived for shallow ones in (Baldassi et al., 2016). Can the authors provide a more detailed explanation?

Empirical novelty and significance:
The second numerical experiment (on continual learning) provides empirical justification to use the proposed algorithm to avoid catastrophic forgetting. The downside of the numerical results is that, again, the neural network architectures that are considered are not very deep. They are wide, but this wasn't as emphasized in the text. In the first numerical experiment, the proposed method does not have any advantages with respect to SGD other than a lower training error. The authors mention that this could suggest that their algorithm leads to better NN capacity, but this is not explored any further in the text. I suggest running simulations with deeper neural networks and expanding on the advantages and disadvantages of the proposed BP-scheme.

Other concerns:
- What is the quality of the approximation in eq. (3), i.e., how much is lost by making this approximation?
- It would be helpful to have a formal definition for the variables in eqs. (6) and (7).
- In the caption of Fig. 2, the authors state that "the training hyperparameters in the two cases are independently selected and generally differ". How are they selected? By cross-validation?




**Summary Of The Paper:**

This paper introduces a belief-propagation message-passing training algorithm for multi-layer neural networks. This algorithm is adapted to mini-batch training and biases distributions toward high entropy solutions. Empirical results show that neural networks with discrete weights and activations trained with this algorithm achieve comparable performance the same networks trained with SGD (BinaryNet), and can make approximate Bayesian predictions that have higher accuracy than pointwise solutions.

**Summary Of The Review:**

Although the paper is a close extension of (Baldassi et al. 2016), the idea to use belief propagation to estimate the marginals of the weights of a neural network (NN) in order to make probabilistic predictions is a novel paradigm and the empirical results on continual learning seem promising. I recommend acceptance subject to the clarification of the concerns discussed in the main review.

---

> ### Author Response · Authors · 2021-11-24
> **Answer to Reviewer 4 - Part 1**
>
>
> We thank the referee for these questions that give us the opportunity to clarify our presentation. Please refer to the general comments to all reviewers for an outline of the additions in the revised paper and for remarks on the significance and novelty of our contribution.
>
> > *it is unclear how the results derived for deep neural networks differ from those derived for shallow ones in (Baldassi et al., 2016). Can the authors provide a more detailed explanation?*
>
> Our BP equations are similar to those of Baldassi et al. 2016, although much more complicated by the fact that they deal with arbitrary deep networks (e.g. we have an additional set of messages and propagation rules). In the revised Appendix A, we now provide a detailed derivation of the message passing algorithms. One further thing to mention is
> that the AMP algorithm is not provided by Baldassi et al. 2016 even on shallow networks, and that the PasP scheme for handling mini-batch updates is also entirely new.
>
> >  *the proposed method does not have any advantages with respect to SGD other than a lower training error.*
>
> We remand the referee to the general comment to all referees for considerations on empirical novelty and relevance. In particular to
> *"We further mention that the purpose of this work is to introduce a novel computational scheme. We felt that showing GD-like performance was sufficient for this purpose. This paper reports for the first time a message passing algorithm capable of training multi-layer neural networks with satisfactory performance. We think that providing a viable alternative to SGD has great scientific value and can potentially generate many directions for future research and applications. Message passing schemes have been shown in the past and in simpler models to be able to achieve algorithmically optimal performance and asymptotic exactness in estimating marginals and entropies. They are amenable to statistical analysis and are able to probe the geometry of the loss landscape. Moreover, we already showed two settings, continual learning and sparse networks, where our scheme outperforms SGD without any further tweaking, and the accuracy of locally Bayesian predictions is enhanced with respect to the point-wise solutions."*
>
> > *The authors mention that [a lower training error] could suggest that their algorithm leads to better NN capacity, but this is not explored any further in the text. [...] [I suggest] expanding on the advantages and disadvantages of the proposed BP-scheme.*
>
> We amended that sentence, it now reads:*"We report that message passing algorithms are able to solve these optimization problems with generalization performance comparable to or better than  SGD-based algorithms.   Some  of  the message passing algorithms (BP and AMP in particular) need fewer epochs to achieve low error than the ones required by SGD-based algorithms, even if adaptive methods like Adam are considered."*. We also outline in the numerical experiments section the increased performance of message passing in the continual learning and sparse layers settings. In the conclusions, we now mention the need for more experimentation and for the extension of our framework to handle biases, batch normalization, and most importantly, convolutional layers.
>
> > *I suggest running simulations with deeper neural networks*
>
> In our original experiments, we trained multi-layer perceptrons with up to 4 layers of weights. These are generally considered to be deep networks: at odds with convolutional and transformer architectures, which also exploit residual connections, adding more layers than that rarely results in performance increase and it is therefore rarely done in practice. Shallow networks typically correspond to 1 or 2 layers MLPs. See e.g. the BinaryNet paper in which the fully connected architecture on the MNIST dataset trains 4 layers of weights (3 hidden neuron layers). In any case, we have conducted experiments on the multiclass Fashion-MNIST dataset training up to 5 layers of weights, verifying the effectiveness of the BP-based algorithms also in this setting.
>
> > *What is the quality of the approximation in eq. (3), i.e., how much is lost by making this approximation?*
>
> The approximation we make is also typical of variational inference approaches: the complex and intractable posterior over the weights is approximated by a factorized distribution, where each marginal is parameterized by a few moments. We are not sure about how much is lost in the approximation (probably a lot) because at that scale is currently not possible to compute a much more accurate posterior.

---

> ### Author Response · Authors · 2021-11-24
> **Answer to Reviewer 4 - Part 2**
>
> > *It would be helpful to have a formal definition for the variables in eqs. (6) and (7).*
>
> The revised Appendix A contains a full derivation of the message passing algorithms where each quantity is properly defined and has an intuitive meaning.
>
> > *In the caption of Fig. 2, the authors state that "the training hyperparameters in the two cases are independently selected and generally differ". How are they selected? By cross-validation?*
>
> We have added the hyper-parameters used for the Bayesian case in Appendix B.3 and a reference in the caption.
> In general, we select hyper-parameters by grid search and choose the ones that yield better performance.

---

### Official Review · Reviewer_nNGL · 2021-11-02

**Correctness:** 3
**Technical Novelty And Significance:** 3
**Empirical Novelty And Significance:** 2
**Recommendation:** 5
**Confidence:** 4

**Main Review:**

**Strengths of the paper**
-  A new paradigm of deep neural network training is prosed based on well-known belief propagation, which is a very interesting and positive try. I do think that such kind of exploration itself is meaningful for future research.
- The resultant algorithms achieve comparable performances as the SGD based methods. In particular, the mini-batch implementation running time is also about the same order as SGD, although apparently, it is still slower than the standard SGD algorithms.

**Weaknesses of the paper**
-  **The first weakness is that there seem no apparent advantages of the performances of the BP-based methods** (even for most complicated BP/BP-I), both in terms of the generalization error and running time (or implementation simplicity). Given this fact, then, one might doubt the practical usage of BP-based methods. As a result, it would be really helpful if the authors could find some scenarios that BP-based methods that are favorable.

 Indeed, it is noticed that the authors had already made an effort in this direction, i.e., the evaluation of local Bayes error and local energy, and the application in continual learning,  similar to the Bayesian neural networks. Some comments are as follows:

1. Using more standard benchmark metrics to quantify uncertainty under distributional shift

Although local Bayes error and local energy are interesting merits, other standard benchmark metrics to quantify uncertainty under distributional shift such as Expected Calibration Error (ECE) and out of distribution (OOD) entropy might be of more interest [1]. It is thus suggested to add experiments on such standard benchmark metrics [1] and compare with other Bayesian deep learning methods.

2. Adding comparisons with previous methods for continual learning (CL)

Although the BP-based method is shown to be well-adapted to continual learning,  this might not be viewed as a special advantage since there are a variety of simple methods for the SGD-based method to enable continual learning [2,3].  All kinds of Bayesian methods (including BP-based) such as variational Bayes, Laplace method et al, are all naturally well-adapted to continual learning within the variational continual learning (VCL) framework[4] and others.  In particular, for binary neural networks considered in this manuscript, there also have been some studies on SGD-based Bayesian training algorithms well-adapted to continual learning, e.g.,[5]. The authors only compare BPI with standard BinaryNet and it is suggested to add some comparison with these previous continual learning algorithms to see if there is any improvement of BP-based methods.

-  **The second weakness is that only binary neural network is considered and there is a lack of evaluation of the continuous weights of BP-based methods**. From my own understanding, the proposed BP-based methods are readily applicable to deal with continuous weights, e.g., AMP was firstly designed to deal with continuous weights. So, why continuous weights are not considered here? It would be helpful if several results for continuous weights can be added with a comparison with SGD based methods.  Otherwise, please add some discussions of the differences between continuous weights and binary weights.

- **The third weakness is that there is a lack of comparisons with several closely related works, in particular, the Expectation Backpropagation (EBP) algorithm [6]**

1. Several closely related works on previous attempts in training deep neural networks using BP-like algorithms are missing. In particular, **[6] proposed one Expectation Backpropagation (EBP) algorithm**, which is applicable to **train deep neural networks with not only binary weights but also continuous weights**. EBP also has a forward pass and backward pass as the paradigm in the current manuscript. Moreover, **EBP has been shown to have similar efficiency and update form as the SGD based method, while achieving competitive performances**.  Although it is based on EP rather than BP in this manuscript, the two are very closely related [7]. In particular, for the AMP variant considered in the current manuscript, **AMP has been proved to be an approximation of EP in [8,9]**. Then, does it imply that the proposed AMP training version is similar to EBP in [6]?  Given such similarity, it is highly suggested to discuss the differences and intrinsic relationships between EBP with the proposed BP-based method, as well as adding comparisons in the experients.

2. Another related method is the proximal mean-field (PMF) method in [10], which is also one training algorithm for deep binary neural networks. **PMF  is proved to be equivalent to a proximal version of the mean-field (MF) method**. As a result, what is the difference between the MF version in the current manuscript from the PMF method in [10]?

3. Moreover, in the Bayesian deep learning community, there are various Bayesian training algorithms using mean-field variational inference methods (eg., [11,12,13]) but are based on SGD methods. Then, what is the key difference between the MF algorithm in the current manuscript with those MF-VI methods? It seems that they (including AMP, BPI, BP) optimize the same variational objective (ELBO bound) but the only difference is the specific optimization methods, previous ones (e.g.,[11,12, 13]) used the SGD to optimize the ELBO bound, while the MF in current manuscript used message passing updates. If so, then maybe the results are expected to be similar or the same? And possibly the BP-based method is not fundamentally different from the SGD method as stated?  Another interesting point is that, from current experimental results, BPI and AMP are basically the same as MF, while apparently BPI and AMP use more accurate (tree-structured, or Bethe) approximations. Can the authors provide some insights into the negligible differences between them?


**Additional Technical Comments**

1. It seems that the so-called Posterior-As-Updates (PasP) is simply the core of the Bayesian theorem in the sequential updates setting (here different mini-batch corresponds to different observations, though possibly with an overlap), which I think is also used in [7] similarly.

2. In the experiment parts, e.g, Figure 1 and Table I, why results of BP are missing?

3. After training, how to perform the prediction using the results of BP-based methods? Are they obtained by sampling from the posterior distribution first and then computing the average output of different samples? Please illustrate clearly in the main text. Also, can the authors plot the posterior distribution of the learned weights?

4. Similarly, how the continual learning is performed? Is it using the learned posterior of the previous task as prior over the next task to learn, similarly as [4,5]?

5. Why the training error of BP-based methods are much lower than SGD while the test error is high or about the same? Is there any intuitive explanation?

6. It is unclear of the so-called _additional reinforcement message_ and how the resultant BP-based algorithms are different from the original versions, e.g., MF, AMP. It would be better to give some explanations in the main text if this point is important.

7. The submitted code of this manuscript seems unavailable.

8. It is suggested to add a discussion of the limitations of the BP-based methods from current results. It seems that although it is comparable (or slightly worse with slightly higher complexity) to SGD methods, additional advantages (Bayes, continual learning) are not apparent either, given that a variety of Bayesian deep learning methods can do the same in a (presumably) simpler way.


**References**

[1] Ovadia, Yaniv, et al. "Can you trust your model's uncertainty? Evaluating predictive uncertainty under dataset shift." NeurIPS 2019.

[2]  Zenke, Friedemann, Ben Poole, and Surya Ganguli. "Continual learning through synaptic intelligence." ICML, 2017.

[3] Parisi, German I., et al. "Continual lifelong learning with neural networks: A review." Neural Networks 113 (2019): 54-71.

[4] Nguyen C V, Li Y, Bui T D, et al. "Variational continual learning." ICLR 2018.

[5] Meng X, Bachmann R, Khan M E. "Training binary neural networks using the bayesian learning rule". ICML 2020.

[6] Soudry, Daniel, Itay Hubara, and Ron Meir. "Expectation backpropagation: Parameter-free training of multilayer neural networks with continuous or discrete weights." NeurIPS. Vol. 1. 2014.

[7] Minka, Tom. Divergence measures and message passing. Technical report, Microsoft Research, 2005.

[8] Meng X, Wu S, Kuang L, et al. "An expectation propagation perspective on approximate message passing". IEEE Signal Processing Letters, 2015, 22(8): 1194-1197.

[9]  B. Cakmak, O. Winther, and B. H. Fleury, “S-AMP: Approximatemessage passing for general matrix ensembles,” in IEEE Information Theory Workshop (ITW), Nov. 2014, pp. 192–196.

[10] Ajanthan T, Dokania P K, Hartley R, et al. "Proximal mean-field for neural network quantization[C]//Proceedings of the IEEE/CVF International Conference on Computer Vision. 2019: 4871-4880."

[11] Alex Graves. "Practical Variational Inference for Neural Networks". In NIPS, 2011

[12] Charles Blundell, Julien Cornebise, Koray Kavukcuoglu, and Daan Wierstra. Weight Uncertainty in Neural Networks. In ICML, 2015.

[13] Osawa, Kazuki, et al.. "Practical Deep Learning with Bayesian Principles". In NeurIPS, 2019.



**Summary Of The Paper:**

This manuscript provides an interesting try on alterative training algorithms for deep neural networks, based on (approximate) message-passing algorithms based on the well-known belief propagation (BP) algorithm.  In particular, the binary neural network is considered and four algorithms (BP, three variants of BP, i.e., BPI, MF, AMP) are proposed within a unified Posterior-As-Prior Update framework. Experiments are conducted on standard supervised classification tasks and continual learning settings, which shows comparable performances as standard SGD based methods.

==========================
After rebuttal:
=========================

I have read the authors' feedback (many thanks for the detailed point-to-point feedback) and other reviewers' comments and modified the score accordingly. Overall, the proposed scheme is interesting, though strictly speaking the results are not very advantageous (at least from its current results) compared to traditional ones, and some of the comparisons seem not very reasonable/fair.



**Summary Of The Review:**

This paper applies the well-known belief propagation (BP) algorithm and several variants to train deep neural networks with binary weights. Overall I like the topic of this manuscript and it is a good start to explore the potentials of BP-based methods for deep learning. While this is indeed a very interesting try, there are several aspects to be improved for the current manuscript, such as a systematic evaluation of the potential advantages of the proposed BP-based methods, a comparison with previous closely related algorithms especially the EBP algorithm and other Bayesian deep learning methods, as well as clarifications of some related technical points, as detailed above. In addition, given the comparable (or slightly worse performance with higher complexity) but not competitive results of the proposed BP-based algorithms, it is worth a discussion of the intuitive reason behind it. This is not saying that it is not useful to study an algorithm without SOTA performance, but rather on the opposite, it is very useful to explain the (intuitive) underlying reason why it works this way even it is not SOTA, as well as its limitations.

---

> ### Author Response · Authors · 2021-11-24
> **Answer Referee 3 - Part 1**
>
> We thank the referee for the detailed review, we find the comments really insightful. Please refer to the general comments to all reviewers for an outline of the additions in the revised paper and for remarks on the significance and novelty of our contribution. We think the low score does not reflect the quality of our work, and try to address below the specific comments raised.
>
> > *The first weakness is that there seem no apparent advantages of the performances of the BP-based methods [...] As a result, it would be really helpful if the authors could find some scenarios that BP-based methods that are favorable.*
>
> We have clarified in the introduction and across the manuscript the novel contributions of our work. In particular ”This is the first work that shows that learning by message passing in deep neural networks 1) is possible and 2) is a viable alternative to SGD” and ”We also remark that our PasP update scheme is of independent interest and can be combined with different posterior approximation techniques.” among others. We stress that while providing for the first time a message passing algorithm able to train deep neural network, we already find in this first work 2 settings, continual learning and sparse layers, where our framework outperforms SGD. Similar considerations are expressed in our general comment to all reviewers.
>
> > *1. Using more standard benchmark metrics to quantify uncertainty under distributional shift*
>
> We thank the referee for the suggestion. We measured the ECE and Brier score as a function of the dataset shift intensity (along the lines of [1]) for the Bayesian predictions of BP and EBP for some of the architectures presented in the main text. In the experiments we have run we found that both the Brier score and the ECE were consistent with the accuracy degradation as the dataset is shifted.
> The results were slightly favorable to BP as expected,
> as we think in this setting the insights given by these measures are well captured by the local energy profiles.
> Unfortunately, due to lack of time, we have not been able to perform an exhaustive set of experiments and we decided not to include these additional results in the text.
>
> > *2. Adding comparisons with previous methods for continual learning (CL)*
>
> While all the suggested methods were specifically designed to avoid catastrophic forgetting, our aim was to characterize our new scheme as it is, and so we thought that the most meaningful comparison would be against a plain version of SGD. Since we explicitly target wide minima, we expected better performances on standard continual learning tasks with respect to standard SGD. We do believe that a specific modification of our method to challenge current SOTA algorithms on the continual learning scenario would be a very interesting direction of research, but we think that this is out of the scope of the current work. We believe that some of the approaches to continual learning the referee mentioned can be adapted to our framework.
>
> > *The second weakness is that only binary neural network is considered and there is a lack of evaluation of the continuous weights of BP-based methods*
>
> We agree on the relevance of the continuous case. The reason for choosing binary weights as the first application is that it represents a simpler case from the point of view of its geometric definition and, at the same time, it is known to be particularly algorithmically challenging. However, we agree with the observation and proceeded to make the extension to the continuous case and include in the revised version (together with a comparison with SGD based methods in the new Fig.2). Needless to say that the binary case has some interest in itself, e.g. for memory and compute constrained devices. However, we do not address these issues here.
>
> > *The third weakness is that there is a lack of comparisons with several closely related works, in particular, the Expectation Backpropagation (EBP) algorithm*
>
> We added a comparison with the EBP algorithm in the text and in the figures.
> We re-implemented the EBP code along the lines of the original matlab implementation.
> We run EBP on the same datasets and architectures presented in our manuscript. The results were quite poor, especially for the point-wise estimator (deterministic output in the original EBP paper). Moreover, (as also mentioned in https://arxiv.org/pdf/1503.03562.pdf) the EBP performances appear to degrade as the network size is increased. In all the tests that we performed, our message passing outperforms the EBP algorithm both in the deterministic and probabilistic predictions (as can be seen in Fig. 2 of the revised manuscript).
> In fact, we did not find published results of EBP performances on deep MLP trained on standard image classification datasets.

---

> > ### Comment · Reviewer_nNGL · 2021-11-29
> > **Many thanks for the feedback**
> >
> > Thank the authors for the detailed point-by-point feedback on my previous comments, which clarifies some confusion. I have raised my score accordingly in recognition of the improvements in the revised version. Nevertheless, still there are several concerns with the current results and presentation, including the advantages of the proposed algorithm and practical usage in real applications. Another main concern, also pointed out by other reviewers, is that the considered deep neural network is not that deep with only two hidden layers.
> >
> > > The results were slightly favorable to BP as expected, as we think in this setting the insights given by these measures are well captured by the local energy profiles. Unfortunately, due to lack of time, we have not been able to perform an exhaustive set of experiments and we decided not to include these additional results in the text.
> >
> > Thanks for the try. It is suggested to add some experiments if time is available in the future version.
> >
> > > While all the suggested methods were specifically designed to avoid catastrophic forgetting, our aim was to characterize our new scheme as it is, and so we thought that the most meaningful comparison would be against a plain version of SGD. Since we explicitly target wide minima, we expected better performances on standard continual learning tasks with respect to standard SGD. We do believe that a specific modification of our method to challenge current SOTA algorithms on the continual learning scenario would be a very interesting direction of research, but we think that this is out of the scope of the current work. We believe that some of the approaches to continual learning the referee mentioned can be adapted to our framework.
> >
> > I still feel that the comparison of message passing with the plain version of SGD in claiming its advantage in continual learning is not very reasonable or fair.  The main reason is that for the message passing algorithm, as explained by the authors, "When switching task, the new prior is the posterior from the last step as usual", which means that a prior distribution is used as regularization to avoid forgetting old tasks in learning new tasks. This is a natural choice in the case of continual learning for any Bayesian approach, like previous Bayesian methods for both binary and continuous neural networks without specific design. In other words, this is an advantage of Bayesian approach, rather than message passing algorithms. In contrast, for naive deterministic algorithms like plain SGD, no such prior regularization is used so that definitely it would be worse than message passing. In a similar spirit of adding a prior, a regularization term could be added to SGD like methods, as is often the strategy in continual learning.
> >
> > > We add that while (MF-)VI is a well-developed and fruitful line of research, message passing methods have a long tradition as well and in simpler systems have been shown to yield exact marginal and be amenable to theoretical description, therefore we think it is valuable to extend these methods to deep learning scenarios.
> >
> > While it is true that message passing algorithms are amenable to the theoretical description for very simple systems (only under certain assumptions), this is not the case for deep neural networks so that this motivation seems not very convincing.
> >
> >
> > >  We now mention in the conclusions a few limitations of our work (e.g. biases, batch normalization, convolutional layers, exact posterior computation).
> >
> > Compared to SGD, during the training procedure, one needs to store much more parameters, e.g., at least doubled since both the mean and variance parameters are stored in message-passing algorithms.
> >
> >
> > > We added a comparison with the EBP algorithm in the text and in the figures...We run EBP on the same datasets and architectures presented in our manuscript. The results were quite poor, especially for the point-wise estimator (deterministic output in the original EBP paper).
> >
> > Many thanks for adding the comparison, which is very useful. Simply curious what happens for the same datasets and architectures in the EBP paper, as it is claimed that they apparently outperform the SGD methods.
> >
> >
> > >  Moreover, we already showed two settings, continual learning, and sparse networks, where our scheme outperforms SGD without any further tweaking.
> >
> > As discussed above, the advantage is continual learning is not very convincing. For the sparse networks, in figure 1, why MF is apparently better than AMP? This brings out another concern that the results of different message passing algorithms are not very consistent.
> >
> > > While the inner loop is implemented as BP iterations the PasP update (outer loop) accumulates messages across different mini-batches, thereby implementing a form of reinforcement/focusing.
> >
> > How about the performance in the full-batch case? It would be helpful to see the effect of reinforcement by adding a full-batch result.

---

> > > ### Author Response · Authors · 2021-11-29
> > > **Answer to the answer to the feedback!**
> > >
> > > We thank the reviewer for her/his stimulating follow-up.
> > >
> > > > still there are several concerns with the current results and presentation, including the advantages of the proposed algorithm and practical usage in real applications.
> > >
> > > In this regard, we would like to reiterate that the purpose of the work was to provide a proof of principle of the efficiency of message passing algorithms. The results are extremely encouraging. Further optimizations of the message passing algorithms are possible and are expected to become available in the future in more applied works.
> > >
> > > > Another main concern, also pointed out by other reviewers, is that the considered deep neural network is not that deep with only two hidden layers.
> > >
> > > As we already commented, already in the original version we showed results with 3 hidden neuron layers, corresponding to 4 trained layers of weights. These are commonly considered deep networks, and deeper MLPs are rarely used in practice.
> > >
> > > > In other words, this is an advantage of Bayesian approach, rather than message passing algorithms.
> > >
> > > We partially agree on this point but also remark that Bayesian algorithms typically require much more epochs to train and perform worse in terms of accuracy on the standard classification task compared to maximum likelihood training. With our approach instead, we perform on par and with fewer epochs on the standard task and better than SGD on the continual task. At the very least, our experiments and comparisons on the standard and the continual tasks show that our approach has some of the advantages of both the Bayesian and the maximum likelihood approach. This should be due to the fact that flat minima are well adapted to switching tasks as noticed in Ref. [1].
> > > We honestly think that more refined comparisons on the continual learning task are beyond the scope of this first paper.
> > >
> > > [1] Feng, Yu and Tu, Yuhai 2021. "The inverse variance-flatness relation in stochastic gradient descent is critical for finding flat minima"
> > >
> > > > While it is true that message passing algorithms are amenable to the theoretical description for very simple systems (only under certain assumptions), this is not the case for deep neural networks so that this motivation seems not very convincing.
> > >
> > > We think that MLPs (+ some simplifying statistical assumptions on the datasets) represent a setting not so out of reach for the analytical techniques, e.g. from statistical physics (~5-10 years?). Our paper sets an algorithm reference that can motivate future studies.
> > >
> > > > Compared to SGD, during the training procedure, one needs to store much more parameters, e.g., at least doubled since both the mean and variance parameters are stored in message-passing algorithms.
> > >
> > > We totally agree, we will mention it in the future revision.
> > >
> > > > For the sparse networks, in figure 1, why MF is apparently better than AMP? This brings out another concern that the results of different message-passing algorithms are not very consistent.
> > >
> > > The study of the sparse case has been done in the short period of the rebuttal, with essentially zero hyperparameter tuning.
> > > Further studies are needed to understand the dependence of AMP parameter choice on sparse graphs. Also, AMP is known to be sensitive to the details of the update scheme.

---

> ### Author Response · Authors · 2021-11-24
> **Answer to Referee 3 - Part 2**
>
>
> > *1. Several closely related works on previous attempts in training deep neural networks using BP-like algorithms are missing. In particular, [6] proposed one Expectation Backpropagation (EBP) algorithm [...] AMP has been proved to be an approximation of EP in [8,9]. Then, does it imply that the proposed AMP training version is similar to EBP in [6]? Given such similarity, it is highly suggested to discuss the differences and intrinsic relationships between EBP with the proposed BP-based method, as well as adding comparisons in the experients.*
>
> EBP can be understood as a cruder version of our message passing algorithms in which rho=1 and 1 inner iteration is performed. The forward pass seems to be the same (for the first inner iteration) but the backward pass differs in an opaque way that would require careful matching of the corresponding terms to be better understood. As EBP is obtained from a first-order Taylor expansion, we suspect that keeping also the diagonal terms of the second-order expansion instead would lead to something more closely related to our algorithms (likely MF or even AMP).
>
> > *2. Another related method is the proximal mean-field (PMF) method [...] what is the difference between the MF version in the current manuscript from the PMF method in [10]?*
>
> The PMF algorithm (exclusively targeting discrete networks training, unlike our approach) can also be obtained from a linearization of a variational objective as the authors show in section 3. Since we do not perform such linearization we do not think the two methods are directly related.
>
> > *3. Moreover, in the Bayesian deep learning community, there are various Bayesian training algorithms using mean-field variational inference methods [...] what is the key difference between the MF algorithm in the current manuscript with those MF-VI methods? [...] previous ones (e.g.,[11,12, 13]) used the SGD to optimize the ELBO bound, while the MF in current manuscript used message passing updates. If so, then maybe the results are expected to be similar or the same? And possibly the BP-based method is not fundamentally different from the SGD method as stated? [...] Can the authors provide some insights into the negligible differences between them [BP, BPI, AMP, MF]?*
>
> The first difference between our approach and VI is that we don't try to optimize the ELBO objective and we don't try to capture the standard posterior. In fact, in our PasP scheme we train a distribution whose mass concentrates during the dynamic on the flat minima. Our training is typically quite fast in terms of number of epochs and both pointwise and (locally) Bayesian predictions are good. On the other hand, MF-VI training typically takes a large number of epochs, is sensitive to the choice of the prior distribution, and the final accuracy is not on par with that of the standard maximum likelihood approach. Bayesian neural networks trained by VI are not used in practice when raw accuracy is needed.
>
> At the inner level, when approximating a mini-batch posterior, MF-VI could provide better results than MF message passing, which suffers converge problem. This is a line of research that we intend to pursue. In principle, MF and MF-VI could admit the same fixed point. BP and AMP are approximations of a different nature instead, they don't optimize the ELBO objective but their fixed points are stationary points of the so-called ELBO free energy.
> We add that while (MF-)VI is a well-developed and fruitful line of research, message passing methods have a long tradition as well and in simpler systems have been shown to yield exact marginal and be amenable to theoretical description, therefore we think it is valuable to extend these methods to deep learning scenarios.

---

> ### Author Response · Authors · 2021-11-25
> **Answer to Referee 3 - Part 3**
>
> On the Additional Technical Comments:
>
> > *1. It seems that the so-called Posterior-As-Updates (PasP) is simply the core of the Bayesian theorem in the sequential updates setting (here different mini-batch corresponds to different observations, though possibly with an overlap), which I think is also used in [7] similarly.*
>
> It would be for \rho=1 and a single pass over the training set. In this case, PasP reduces to the assumed density filtering algorithm as mentioned in our paper.
> When multiple passes are taken instead, in order to obtain an (approximate) Bayesian prediction
> one would need to counterbalance the effect of multiple observations of the same data. This is what the tilted distributions in Expectation Propagation are for, as discussed in [7]. In PasP we don't counterbalance: this leads the approximate posterior to concentrate on highly entropic states as the training goes on.
>
>
> > *2. In the experiment parts, e.g, Figure 1 and Table I, why results of BP are missing?*
>
> BP and BPI are the same if tau_{max}=1, we clarified this in the text, the figure and the tables (referring only to BP and not to BPI).
>
> > *3. After training, how to perform the prediction using the results of BP-based methods? Are they obtained by sampling from the posterior distribution first and then computing the average output of different samples? Please illustrate clearly in the main text. Also, can the authors plot the posterior distribution of the learned weights?*
>
> We obtain pointwise predictions by taking the sign of the posterior means (magnetizations) for binary weights or just the means for the continuous ones. Bayesian predictions are simply obtained from a single forward pass of the message passing. We'll make sure to clarify these points in future versions of the paper.
>
> We have plotted the learned magnetizations from our algorithms for binary networks (yet to be added to the manuscript).  The histogram is broader and more bimodal around -1 and +1 compared to BinaryNet which also presents a larger fraction of weights near zero.
>
> > *4. Similarly, how the continual learning is performed? Is it using the learned posterior of the previous task as prior over the next task to learn, similarly as [4,5]?*
>
> We did not design a specific strategy for the continual learning task. Each task involves multiple PasP steps. When switching task, the new prior is the posterior from the last step as usual.
>
> > *5. Why the training error of BP-based methods are much lower than SGD while the test error is high or about the same? Is there any intuitive explanation?*
>
> Let us first mention that fast convergence is a well-known property of BP-based algorithms, e.g. in decoding and inference. We have also checked (now in the Appendix) that the solutions found by SGD, even if corresponding to slightly higher training error, also belong to wide flat valleys on the error landscape.
> We also notice that there are cases (e.g. over-parameterized nonconvex shallow networks) in which it can be shown that  fBP reaches zero error while SGD does not, and these statements may be made more precise and also studied analytically (https://arxiv.org/abs/2110.00683).
>
> > *6. It is unclear of the so-called additional reinforcement message and how the resultant BP-based algorithms are different from the original versions, e.g., MF, AMP. It would be better to give some explanations in the main text if this point is important.*
>
> Indeed this is a *key point* of the paper. We have rewritten the explanation and hopefully improved its clarity. In particular, we have discussed separately the outer and inner loops of the message passing algorithms. While the inner loop is implemented as BP iterations the PasP update (outer loop) accumulates messages across different mini-batches, thereby implementing a form of reinforcement/focusing.
>
> > *7. The submitted code of this manuscript seems unavailable.*
>
> At publication time, we intend to provide in the main text the link to a public GitHub repository with the code used to perform the experiments.
>
> > *8. It is suggested to add a discussion of the limitations of the BP-based methods from current results.*
>
> We now mention in the conclusions a few limitations of our work (e.g. biases, batch normalization, convolutional layers, exact posterior computation).

---

### Official Review · Reviewer_tYY4 · 2021-11-06

**Correctness:** 3
**Technical Novelty And Significance:** 2
**Empirical Novelty And Significance:** 2
**Recommendation:** 6
**Confidence:** 4

**Main Review:**

[Main Strengths]

This paper's main strength is that the authors' method of using message-passing to train NNs is pretty interesting, the introductory section is well-written, and the implementation GitHub repo is intended to be provided.

=================================================================

[Main Weaknesses]

The fundamental shortcoming of this study is that, while the basic concept is intriguing, it does not appear to make a significant impact, much like the various flavors of SGD. I invite authors to make the further improvement suggested in Section 4.4, which is to show that message passing is inherently less prone to catastrophic forgetting issues, which will be highly intriguing and will undoubtedly require more clear justification.

=================================================================

[Technical Comments]

1) Is it possible to expand this method to multi-class scenarios, such as extending message-passing decoding algorithms from binary linear codes to non-binary codes?
2) Furthermore, substantial message passing successes have occurred in the past, particularly for sparse factor graphs. As a result, future applications of Graph neural networks or sparse Transformers will be quite fascinating.

=================================================================

[Typographical comments]
1) To ensure perfect anonymity, erase the name displayed in Acknowledgement.
2) On page 3, the term "PasP rule" is not defined until it is used in Section 2.


**Summary Of The Paper:**

[Summary]

This paper develops a class of fBP-based message-passing algorithms by adding a
“reinforcement “ term to the BP equations and shows equivalent performance to the binary networks in experiments.

**Summary Of The Review:**

Using message-passing to train NNs is intriguing, but, like the various flavors of SGD, it does not appear to make a significant difference.

---

> ### Author Response · Authors · 2021-11-23
> **Answer to Referee 2**
>
>
> We thank the referee for the positive comments on our work.
> Regarding the [Main Weaknesses], we kindly point the referee to the general comment to all reviewers. We stress that while providing for the first time a message passing algorithm able to train deep neural network, we already find in this first work 2 settings, continual learning and sparse layers, where our framework outperforms SGD.
>
>
> > *Is it possible to expand this method to multi-class scenarios, such as extending message-passing decoding algorithms from binary linear codes to non-binary codes?*
>
> In the revised paper we perform additional experiments with continuous weights. For the case of discrete but non-binary weights instead, we see no problem in accomodating it in our general formalism: the integral in \psi(H,G,\theta) becomes a sum over the discrete alphabet and the distribution q_\theta(w) a discrete distribution.
>
> > *Furthermore, substantial message passing successes have occurred in the past, particularly for sparse factor graphs. As a result, future applications of Graph neural networks or sparse Transformers will be quite fascinating.*
>
> We performed an additional set of experiments along this direction, by imposing a (random) fixed sparsity pattern on each layer' weights. We were happy to find that in this scenario message passing is able to outperform SGD.

---

### Official Review · Reviewer_jTGp · 2021-11-06

**Correctness:** 3
**Technical Novelty And Significance:** 2
**Empirical Novelty And Significance:** 2
**Recommendation:** 3
**Confidence:** 4

**Main Review:**

Clarity of writing:  The manuscript contains a significant number of typos that, while irritating to read, do not inhibit understanding.  The larger issues are the plethora of undefined or underdefined terminology, e.g., channel functions, damping, etc., the imprecise mathematical formulations, e.g., dimension of vectors, z^l, etc., and the lack of motivation for mathematical formulations, e.g., BP is never really defined and equations (8)-(19) are unmotivated.

Novelty:  The general approach (with the prior) seems novel to me, but it is a bit unclear exactly what other pieces are novel here compared to existing work.

Significance:  While it is interesting that a BP style message passing approach can achieve comparable levels of performance to SGD, the overall significance of the works seems somewhat limited.  In particular, I'm not sure that the authors really present a compelling example of when this approach would be preferred over pure SGD based solutions.

Specific comments:

- The introduction makes vague claims without support, e.g., no citations for applications of BP, "additional approximation turns out to be benign", etc.  Consider making it more precise.
- The paper conflates factorizations with factor graphs (which aren't really discussed at all).
- "non-homogeneous" -> inhomogeneous
- What is a scalar channel function?  It needs a definition.
- "The training error is usually lower for optimized configurations from message passing schemes, suggesting that these algorithms are able to achieve higher capacity than SGD-based algorithms (given the same test error)"  While, looking that the plots, the convergence of some of the BP variants is certainly better than SGD, I'm not sure that your statement would continue to hold if you ran for another 100 epochs.
- Do you use adaptive step size methods like ADAM for SGD?
- The experiments make claims about deep neural networks, but most of the experimental results are on shallow networks.
- The number of repeated runs in the experiments (5 in some cases) might be a little small.

**Summary Of The Paper:**

The authors describe a "focused" belief propagation strategy for learning NNs.  They argue that it continues to exhibit some of the same nice properties that stochastic gradient methods have in practice with the added benefit of allowing the computation of approximate marginals to improve the accuracy of predictions.

**Summary Of The Review:**

An interesting approach that is hampered by a poor presentation, somewhat limited experimental results, and only minor justification for why it should be seriously considered as an alternative approach to more well-studied approaches for training neural networks.

---

> ### Author Response · Authors · 2021-11-23
> **Answer to Reviewer 1 - Part 1**
>
> We thank the reviewer for his/her comments. We think that the revised paper + the general comment to all reviewers + the detailed answers below should address the major concern put forward.
>
> > *Clarity of writing: ....*
>
> We agree with the referee that the presentation of the manuscript needed to be improved (disadvantage of writing under deadline). We have implemented a substantial revision in order to make the manuscript more self-contained.
> Besides the changes reported in the general comment to all referees, we have:
> - Expanded the Related Works section, providing more context and motivation for the message passing approaches.
> - Expanded Appendix A to contain a complete set of definitions and the full derivation of the message passing algorithms, instead of relying on external references to fill in the gaps.
> - Added Appendix B.2 with an explanation of the damping procedure.
>
> > *Novelty:  The general approach (with the prior) seems novel to me, but it is a bit unclear exactly what other pieces are novel here compared to existing work.*
>
> We have clarified in the introduction and across the manuscript the novel contributions of our work.  In particular ”This is the first work that shows that learning by message passing in deep neural networks 1) is possible and 2) is a viable alternative to SGD” and ”We also remark that our PasP update scheme is of independent interest and can be combined with different posterior approximation techniques.” among others.
>
> > *Significance: While it is interesting that a BP style message passing approach can achieve comparable levels of performance to SGD, the overall significance of the works seems somewhat limited. In particular, I'm not sure that the authors really present a compelling example of when this approach would be preferred over pure SGD based solutions.*
>
> We kindly point the referee to the general comment to all reviewers for the significativity issue. We stress that while providing for the first time a message passing algorithm able to train deep neural network, we already find in this first work 2 settings, continual learning and sparse layers, where our framework outperforms SGD.
>
> > *The introduction makes vague claims without support, e.g., no citations for applications of BP, "additional approximation turns out to be benign", etc. Consider making it more precise.*
>
> We understand the observation. We intended to help the reader by focusing on the main points while avoiding going into non-essential details. However, we realize that this approach may ultimately be counterproductive. As a result, we have amended the main text and the appendix, clarified every detail as much as possible, trying not to make the reading too heavy.
>
> > *The paper conflates factorizations with factor graphs (which aren't really discussed at all)*
>
> Fixed the conflation and provided a brief explanation of factor graphs in the main text and in Appendix A.

---

> ### Author Response · Authors · 2021-11-23
> **Answer to Reviewer 1 - Part 2**
>
>
> > *What is a scalar channel function? It needs a definition.*
>
> Removed the "channel" nomenclature (borrowed from the signal processing literature) and clarified definition.
>
> > *"The training error is usually lower for optimized configurations from message passing schemes, suggesting that these algorithms are able to achieve higher capacity than SGD-based algorithms (given the same test error)" While, looking that the plots, the convergence of some of the BP variants is certainly better than SGD, I'm not sure that your statement would continue to hold if you ran for another 100 epochs.*
>
> We agree on this point. We have accordingly changed the sentence in the main text to  "Some of the
> message passing algorithms (BP and AMP in particular) need fewer epochs to achieve low error than the ones required by SGD-based algorithms, even if adaptive methods like Adam are considered."
>
> > *Do you use adaptive step size methods like ADAM for SGD?*
>
> In the original version of the paper, we used standard SGD. We have now added a comparison using the Adam optimizer in the new Fig.2 and used Adam in the right panel of the new Fig.1. We notice that with little hyper-parameters tuning performances are similar among the 2.
>
> > *The experiments make claims about deep neural networks, but most of the experimental results are on shallow networks.*
>
> In our original experiments, we trained multi-layer perceptrons with up to 4 layers of weights. These are generally considered to be deep networks: at odds with convolutional and transformer architectures, which also exploit residual connections,  adding more layers than that rarely results in performance increase and it is therefore rarely done in practice. Shallow networks typically correspond to 1 or 2 layers MLPs. See e.g. the BinaryNet paper in which the fully connected architecture on the MNIST dataset trains 4 layers of weights (3 hidden neuron layers). In any case, we have conducted experiments on the multiclass Fashion-MNIST dataset training up to 5 layers of weights, verifying the effectiveness of the BP-based algorithms also in this setting.
>
> > *The number of repeated runs in the experiments (5 in some cases) might be a little small.*
>
> Given time constraints and the plethora of additional experiments we have performed, we did not have time to systematically augment the number of repeated runs.

---

### Author Response · Authors · 2021-11-19
**Comments to all reviewers**


We thank all the reviewers for their comments, we think they largely helped in improving the paper. Before uploading the revised version of the paper, we list here the major additions:

- **Added experiments with continuous weights.** Although our message passing formalism encompasses both discrete and continuous weights and arbitrary activations, the experiments were done using binary weights and binary activations. Since we propose an alternative to optimization based schemes, we could not leverage existing deep learning frameworks and basically had to implement one from scratch, therefore we initially selected a single setting for simplicity. We now add a set of experiments with continuous weights. We find that the performances of BP-based algorithms with continuous weights are comparable with the ones obtained by SGD with continuous weights, strongly enhancing the applicability of our method. Let us mention that also in this case it is possible to make locally Bayesian predictions that enhance the generalization accuracy compared to SGD.
- **Comparison with Expectation Backpropagation.**
    We tested EBP on the same datasets and architectures presented in our manuscript. The results were quite poor, especially for the point-wise estimator (deterministic output in the original EBP paper).
    In all our tests the message passing approaches outperform the EBP algorithm in both the deterministic and probabilistic predictions.

- **Added experiments with sparse layers.**
    In order to address the concerns by the referees on the practical application and significance of our method, and to find a context in which it presents an advantage in performance over SGD, we experimented with multi-layer perceptrons where a random fixed (during training) sparsity pattern (90% of weights set to 0) is imposed on each layer. This experiment is stimulated by the suggestions of the second referee (tYY4) and by the fact that algorithms based on message passing have historically been successful on sparse graphs.
    *In this setting, our message passing significantly outperforms SGD.* This finding is particularly relevant in light of the vast amount of research in sparsity and pruning in neural networks.

- **Detailed derivation of message passing algorithms.**
   While the reader was previously directed to general references to fill in the gaps in the derivation of the various message passing equations, we have added in the revised version the explicit derivation of these equations. The purpose is to achieve greater clarity and self-consistency in the manuscript.

- **Expanded "Related Works" section.**
    We give a broader overview of the community's difficulties in trying to extend message passing algorithms to the deep neural networks setting. In our paper, we present the first successful scheme to train deep networks through message passing. A crucial ingredient of our scheme is the Posterior-as-Prior update, which is derived as an adaptation to deep networks of the focusing BP (fBP) equations which are designed to efficiently estimate marginals in wide flat minima of the training error loss landscape.

- **Improved exposition and performed requested additional experiments**.
    We have revised exposition and mathematical definitions to improve clarity and self-consistency of our work. On the experimental side, we have in particular added experiments with more layers as pointed out by some of the referees.

We further mention that the purpose of this work is to introduce a novel computational scheme.  We felt that showing  GD-like performance was sufficient for this purpose. This paper reports for the first time a message passing algorithm capable of training multi-layer neural networks with satisfactory performance.
We think that providing a viable alternative to SGD has great scientific value and can potentially generate many directions for future research and applications.
Message passing schemes have been shown in the past and in simpler models to be able to achieve algorithmically optimal performance, asymptotic exactness in estimating marginals and entropies. They are amenable to statistical analysis and are able to probe the geometry of the loss landscape.
Moreover, we already showed two settings, continual learning and sparse networks, where our scheme outperforms SGD without any further tweaking, and the accuracy of locally Bayesian predictions is enhanced with respect to the point-wise solutions.

We hope that these additions and the above clarification address the major concerns of all reviewers and encourage them to increase their scores since we believe they do not reflect the novelty and relevance of our work. We will upload the revised manuscript and punctually answer the reviewers' comments in the coming days.

---

### Decision · Program_Chairs · 2022-01-20

**Decision:**

Reject

**Comment:**

This paper presents a method for training neural networks with belief propagation-based algorithms. The approach is to set a fully factorized prior over weights, compute a forward and backward pass of messages on a minibatch, then set the new prior to be a slightly higher temperature version of the minibatch approximate posterior. This new prior is then used for the next minibatch, and training iterates.

There is a huge range of opinions amongst reviewers. The main thing that reviewers appreciate is the novelty of using belief propagation instead of backpropagation for training neural networks. Finding alternatives to backprop with favorable properties could be hugely impactful, so even small gains in this direction are valuable. The posterior-as-prior update is interesting, and the authors have clearly put in care to getting things working. The main weaknesses are that some of the experiments aren’t always reasonable and fair, the paper is framed to overstate its contribution, and there’s not a clear advantage over standard approaches (e.g., MNIST error rates for a two hidden layer network are >2%).

In the end, this is a very borderline paper, but I find Rev nNGL’s position to be most informative. In particular, the paper frames the main contribution to be message passing as an alternative to SGD for training neural networks, but this is too broad of a framing given the existence of other closely related approaches like Soudry et al pointed out by Rev nNGL. I’d recommend that the authors frame their work as an advance over other message passing-based approaches to training neural networks, and to focus on piecing apart precisely why the proposed approach improves over EBP and alternatives.